# Retinol dehydrogenase 10 reduction mediated retinol metabolism disorder promotes diabetic cardiomyopathy in male mice

Yandi Wu [1,2,8], Tongsheng Huang [1,2,8], Xinghui Li[1,2,8], Conghui Shen[1,2], Honglin Ren[1,2], Haiping Wang[3], Teng Wu[1,2], Xinlu Fu[1,2], Shijie Deng[1,2], Ziqi Feng[1,2], Shijie Xiong[1,2], Hui Li[1], Saifei Gao[1], Zhenyu Yang[1], Fei Gao[4], Lele Dong[4], Jianding Cheng[5,6,9] & Weibin Cai [1,2,7] ✉

Diabetic cardiomyopathy is a primary myocardial injury induced by diabetes with complex pathogenesis. In this study, we identify disordered cardiac retinol metabolism in type 2 diabetic male mice and patients characterized by retinol overload, all-trans retinoic acid deficiency. By supplementing type 2 diabetic male mice with retinol or all-trans retinoic acid, we demonstrate that both cardiac retinol overload and all-trans retinoic acid deficiency promote diabetic cardiomyopathy. Mechanistically, by constructing cardiomyocyte-specific conditional retinol dehydrogenase 10-knockout male mice and over-expressing retinol dehydrogenase 10 in male type 2 diabetic mice via adeno-associated virus, we verify that the reduction in cardiac retinol dehydrogenase 10 is the initiating factor for cardiac retinol metabolism disorder and results in diabetic cardiomyopathy through lipotoxicity and ferroptosis. Therefore, we suggest that the reduction of cardiac retinol dehydrogenase 10 and its mediated disorder of cardiac retinol metabolism is a new mechanism underlying diabetic cardiomyopathy.

Diabetes mellitus (DM) is an independent risk factor for cardiovascular diseases[1] and can independently induce structural and functional disruptions in the heart, leading to diabetic cardiomyopathy (DCM)[1]. With the application of new therapies, most complications of DM have been effectively controlled, and the life expectancy of DM patients has been extended, while hidden myocardial injury, which causes 50–80% of deaths, has become the leading cause of death in DM patients[2–4]. Mechanistic studies are key to improving the prevention and

treatment of DCM. DCM has complex pathogenesis in which lipotoxicity and myocardial cell death should not be underestimated[5,6].

Retinol (vitamin A, Rol) and all-trans retinoic acid (atRA) are metabolites of retinol metabolism that have been shown to have altered levels in diseases[7,8]. The study demonstrated that retinoic acid receptors (RARs) were reduced in the hearts of diabetic rats and activating these receptors by atRA or other activators prevented myocardial injury[9]; however, it remains unknown whether cardiac retinol

[1]Guangdong Engineering & Technology Research Center for Disease-Model Animals, Laboratory Animal Center, Zhongshan School of Medicine, Sun Yat-sen University, Guangzhou 510080 Guangdong, China. [2]Department of Biochemistry, Zhongshan School of Medicine, Sun Yat-sen University, Guangzhou 510080 Guangdong, China. [3]Prenatal Diagnosis Center, Guangdong Provincial People's Hospital, Guangdong Academy of Medical Sciences, Guangzhou 510080 Guangdong, China. [4]Durbrain Medical Laboratory, Hangzhou 310000 Zhejiang, China. [5]Department of Forensic Pathology, Zhongshan School of Medicine, Sun Yat-sen University, Guangzhou 510080 Guangdong, China. [6]Guangdong Province Translational Forensic Medicine Engineering Technology Research Center, Sun Yat-Sen University, Guangzhou 510080 Guangdong, China. [7]Guangdong Provincial Key Laboratory of Digestive Cancer Research, The Seventh Affiliated Hospital of Sun Yat-sen University, Shenzhen 518107 Guangdong, China. [8]These authors contributed equally: Yandi Wu, Tongsheng Huang, Xinghui Li. [9]Deceased: Jianding Cheng. ✉e-mail: caiwb@mail.sysu.edu.cn

metabolite levels are altered and whether these alterations are involved in DCM.

In retinol metabolism, Rol, the metabolic substrate, undergoes a two-step dehydrogenation reaction to generate the active metabolite atRA, which then primarily binds and activates RARs (ligand-induced transcription factors) to exert biological effects[10]. Retinol dehydrogenase 10 (RDH10) is the rate-limiting enzyme in retinol metabolism that has an important role in the conversion of Rol to atRA and affects organ development during embryonic development[11,12], but the role of RDH10 in retinol metabolism and disease in adulthood, particularly in the heart, remains poorly understood.

In this study, we demonstrated that in type 2 diabetes mellitus (T2DM), impaired cardiac retinol metabolism caused by cardiac RDH10 reduction results in DCM through Rol overload-induced cardiotoxicity and atRA deficiency-induced lipotoxicity and ferroptosis.

## Results

### Retinol metabolism disorder in the hearts of mice and patients with T2DM

Our preliminary study showed that the *db/db* mouse is not only a T2DM model but also a good DCM model with a typical pattern of heart function changes from a compensatory stage to a decompensatory stage[6]. We selected *db/db* mice at 4, 24, and 32 weeks of age and performed RNA-seq analysis of their hearts. The selected mice were identified as prehyperglycemic, hyperglycemic with compensated heart function, and hyperglycemic with decompensated heart function (Supplementary Fig. 1).

There were 1695, 2584, and 2147 differentially expressed genes (DEGs, fold change ≥1.5) identified in the hearts of *db/db* mice at 4, 24, and 32 weeks of age, respectively, of which 643, 1293, and 1153 were upregulated while 1052, 1291 and 994 were downregulated (Fig. 1a). A total of 215 DEGs were changed in all 3 groups (Fig. 1b) and were significantly enriched ($p < 0.001$, $q < 0.001$) in 14 pathways classified by Kyoto Encyclopedia of Genes and Genomes (KEGG), with metabolic pathways (mmu01100) containing the most genes (Fig. 1c). Gene ontology (GO) classification of the DEGs in metabolic pathways was then performed and found that most of these DEGs were significantly enriched ($p < 0.001$, $q < 0.001$) in the response to vitamin A (GO:0033189), retinoic acid metabolic process (GO:0042573) and retinol metabolic process (GO:0042572) (Fig. 1d).

Further analysis of DEGs in retinol metabolism (mmu00830) showed that retinoic acid synthesis-related genes, *ALDH1A2*[13] and *ALDH1A7*[14,15], were gradually downregulated, while retinoic acid degradation-related genes, *CYP3A11*[16], *CYP26B1*[13,17], *UGT1A6B*[18] and *UGT1A6A*[18,19], were gradually upregulated (Fig. 1e). More importantly, we found a nearly 1-fold increase in Rol, a halving of atRA, and significant decreases in retinoic acid receptor α (RARa) and retinoic acid receptor β (RARb) in the hearts of 32-week-old *db/db* mice (Fig. 1f and g), indicating a retinol metabolism disorder characterized by Rol overload, atRA deficiency, and RARs reduction was present in the hearts of T2DM mice. The reduction of RARa and RARb was also found in the hearts of T2DM patients, suggesting the presence of cardiac retinol metabolism disorder in T2DM patients (Fig. 1h).

### Rol aggravated cardiac Rol overload and exacerbated myocardial injury by promoting myocardial fibrosis and apoptosis in T2DM mice

To demonstrate the role of Rol overload in DCM, we supplemented *db/db* mice with Rol. We defined an experimental baseline and started Rol supplementation at 8 weeks of age when *db/db* mice developed stable hyperglycemia and measured body weight, blood glucose, and heart function over the following period until week 24 after baseline, when Rol-supplemented *db/db* mice developed heart failure manifested by a significant decrease in left ventricular ejection fraction (LVEF) and left

ventricular fractional shortening (LVFS) (Fig. 2a–c). Cardiac Rol and atRA levels and cardiac structural remodeling were also measured at week 24 and the results showed that in *db/db* mice, Rol significantly increased cardiac Rol but not atRA (Fig. 2d) or retinyl ester levels (Supplementary Fig. 2) and exacerbated cardiac structural remodeling including myocardial fibrosis, apoptosis, myofibrillar fragmentation and increased mitochondria in cardiomyocytes (Fig. 2e–g). Additionally, we found that Rol reduced hyperglycemia, hyperinsulinemia, insulin resistance, and to some extent dyslipidemia in *db/db* mice (Fig. 2h–l), suggesting that the effects of Rol on myocardial injury in *db/db* mice were not by exacerbating diabetes. At the same time, considering the increase in cardiac Rol levels, we suggest that it is the overload of cardiac Rol that promotes myocardial injury in T2DM mice.

### atRA restored cardiac atRA levels and alleviated myocardial injury in T2DM mice

To demonstrate the role of atRA deficiency in DCM, we supplemented atRA with *db/db* mice. We defined the experimental baseline and started atRA supplementation in *db/db* mice at 8 weeks of age and measured body weight, blood glucose, and heart function over the following period until week 28 after baseline when *db/db* mice without atRA supplementation developed heart failure (Fig. 3a–c). Cardiac atRA levels and structural remodeling were also measured at week 28 after baseline and the results showed that in *db/db* mice, atRA supplementation restored cardiac atRA levels (Fig. 3d) and attenuated cardiac structural remodeling including myocardial hypertrophy, myocardial fibrosis, apoptosis, lipid deposition and increased mitochondria in cardiomyocytes (Fig. 3e–g). Additionally, atRA reduced hyperinsulinemia, insulin resistance, and to some extent dyslipidemia in *db/db* mice (Fig. 3h–l).

These results suggest that atRA prevented myocardial injury by restoring cardiac atRA levels and improving systemic metabolic disorder in T2DM mice.

### RDH10 expression was reduced in the hearts of mice and patients with T2DM

The levels of cardiac Rol and atRA in untreated and Rol-supplemented *db/db* mice (Fig. 1f and Fig. 2d) suggested that the conversion of Rol to atRA is disrupted in the heart in T2DM. Therefore, we measured the expression of cardiac RDH10, a rate-limiting enzyme in retinol metabolism that plays an important role in converting Rol to atRA, and found that as heart function shifted from compensated to decompensated, cardiac RDH10 expression increased and then decreased in *db/db* mice (Supplementary Fig. 3a, b and d) while we found a significant decrease in the activity of RDHs in the myocardium of *db/db* mice (Supplementary Fig. 3c)A severe reduction in cardiac RDH10 was also found in T2DM patients (Supplementary Fig. 3d). These results suggest that RDH10 may be closely associated with disturbed cardiac retinol metabolism and its resulting DCM in T2DM.

### The loss of cardiac RDH10 led to disordered cardiac retinol metabolism and severe myocardial injury in mice

To further confirm the relationship between RDH10, cardiac retinol metabolism, and myocardial injury, we bred mice with cardiomyocyte-specific RDH10 deletion. Global RDH10 knockout is embryonically lethal[11] and causes abnormal development in several organs, including the heart[20]. To exclude possible effects of abnormal heart development and to investigate the relationship between RDH10, retinol metabolism, and myocardial injury in adult heart more specifically, we used the myh6 system to generate cardiomyocyte-specific conditional RDH10-knockout (RDH10-cKO) mice by tamoxifen (TMX) treatment at 5 weeks of age to induce iCre protein specifically in cardiomyocytes (Fig. 4a) and verified the RDH10 deletion in cardiomyocytes of these mice after 4 weeks

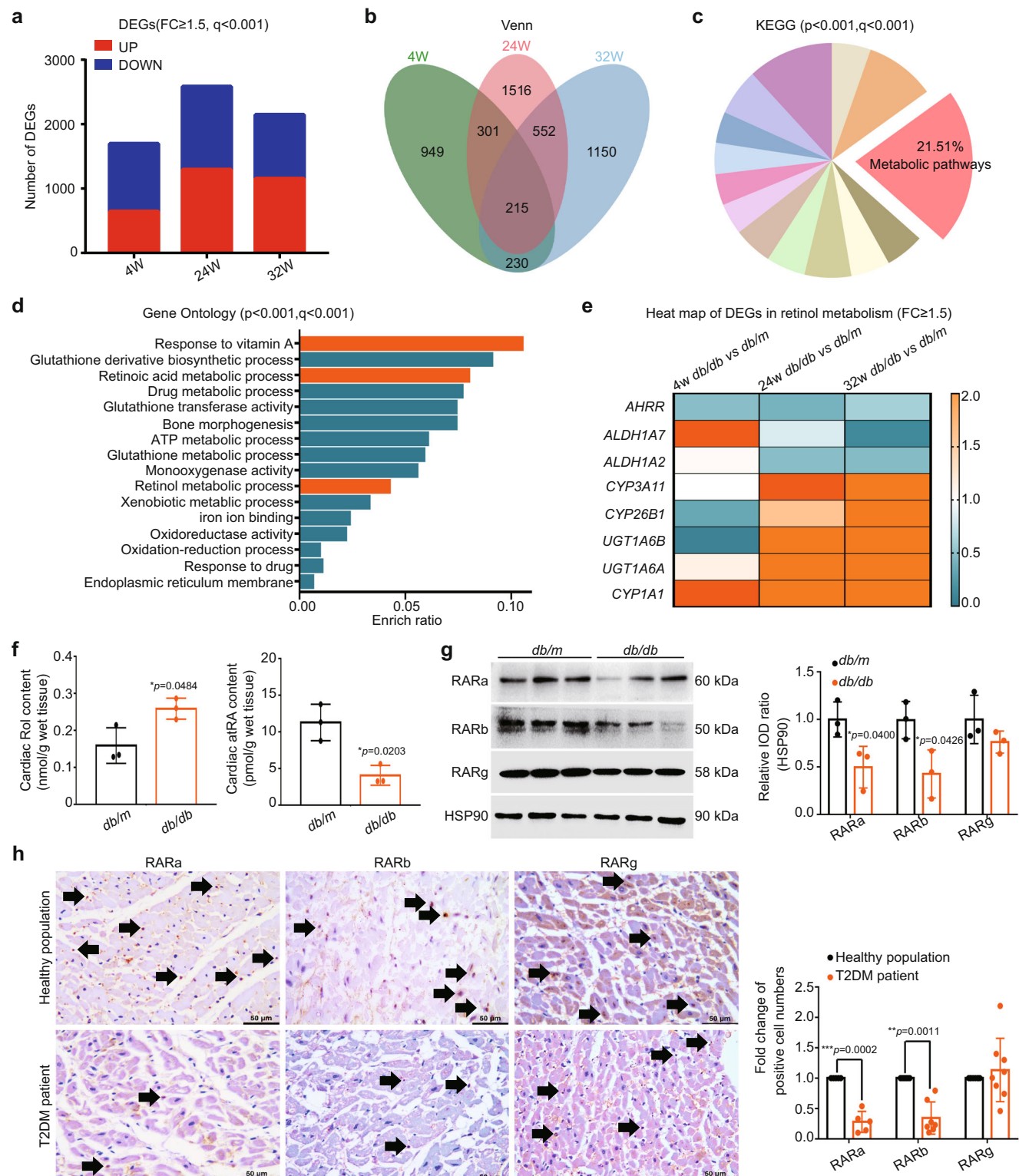

**Fig. 1 | Altered cardiac retinol metabolism in type 2 diabetes mellitus (T2DM) (n means biologically independent animals). a** Summary of differentially expressed genes (DEGs). **b** Venn diagram of DEGs. **c** Kyoto Encyclopedia of Genes and Genomes (KEGG) analysis of DEGs. **d** Gene ontology (GO) terms analysis of DEGs in metabolic pathways. **e** Heat map of DEGs in retinol metabolism pathway. **f** Cardiac retinol (Rol) and all-trans retinoic acid (atRA) levels in *db/db* mice, $n = 3$, * *vs db/m*. **g** Western blotting (WB) of cardiac (retinoic acid receptors) RARs, $n = 3$, * *vs db/m*. **h** Cardiac Immunohistochemistry (IHC) staining of T2DM patients, *n* (retinoic acid receptor a, RARa) = 5 (T2DM patient and healthy population), *n* (retinoic acid receptor b, RARb) = 7 (T2DM patient and healthy population) and *n* (retinoic acid receptor g, RARg) = 8 (T2DM patient and healthy population), * *vs* healthy population. Data are expressed as means ± SD. Two-tailed unpaired *t*-test was used for the analysis of statistical significance. Source data are provided as a Source Data file. (Black arrows: Typical IHC stained positive cells).

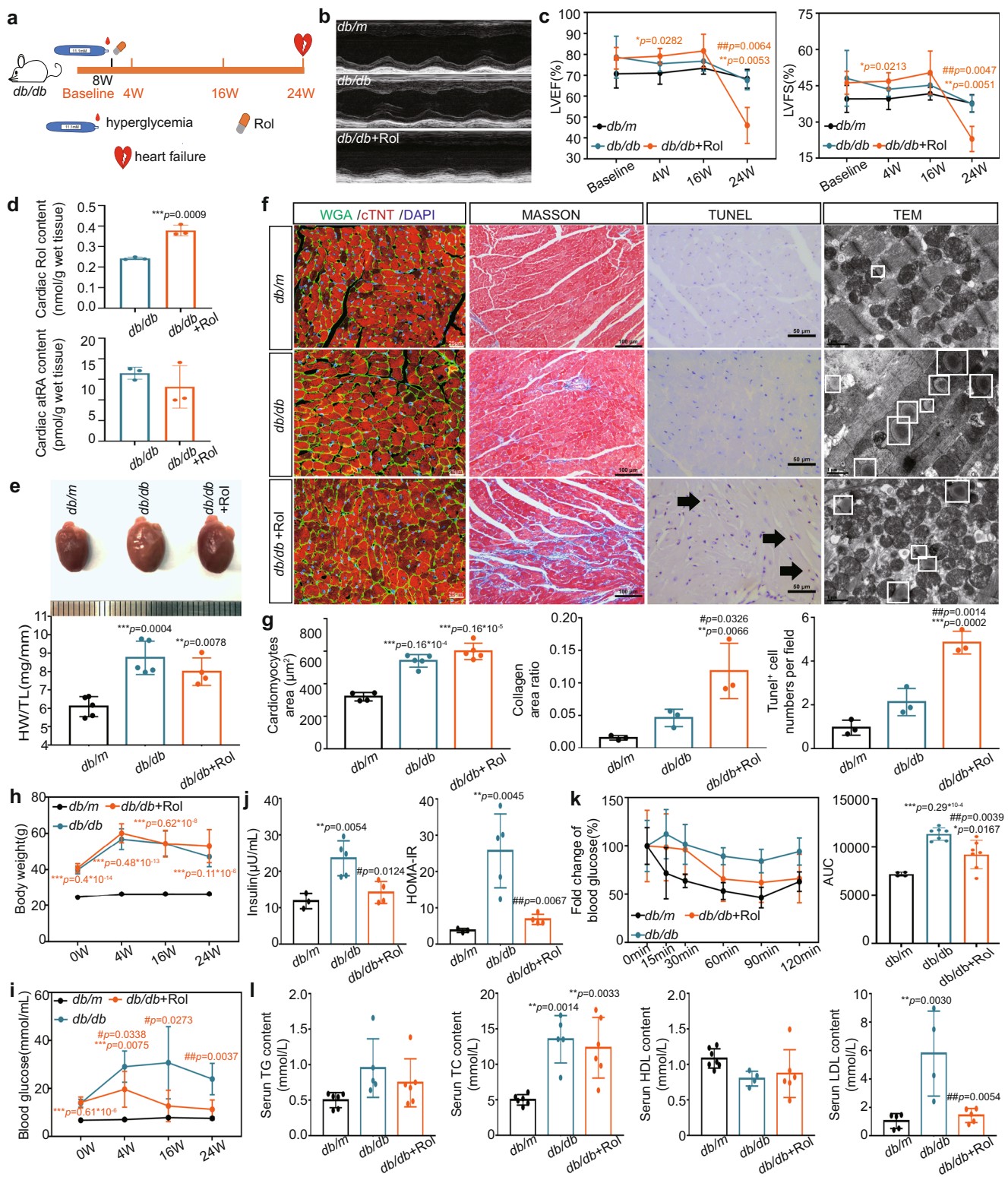

(Supplementary Fig. 4b and c). Our further results showed that at week 15 after TMX treatment, RDH10-cKO mice exhibited a halving of cardiac atRA (Fig. 4b), heart failure (Fig. 4c and d), and severe cardiac structural remodeling including myocardial hypertrophy, myocardial fibrosis, apoptosis, lipid deposition and increased mitochondria in cardiomyocytes (Fig. 4e–g).

These results suggest that the absence of cardiac RDH10 and its resultant cardiac retinol disorder manifested mainly as atRA deficiency can independently induce myocardial injury.

## RDH10 overexpression ameliorated cardiac retinol metabolism disorder and myocardial injury in T2DM mice

To further validate the role of RDH10 and cardiac retinol metabolism in DCM, we overexpressed RDH10 in the hearts of db/db mice via adeno-associated virus 9 (AAV9)-RDH10 injection at week 8 after experimental baseline (Fig. 5a) and verified that the expression of cardiac RDH10 of these mice restored to a level comparable to that of db/m mice after 5 weeks (Supplementary Fig. 5a and b). At 28 weeks of hyperglycemia in db/db mice, AAV9-RDH10 alleviated cardiac retinol

**Fig. 2 | Cardiac retinol metabolic status, structure, and function as well as serum insulin and lipids in T2DM mice supplemented with Rol (*n* means biologically independent animals). a** Schematic diagram of Rol supplementary. **b** Echocardiography. **c** left ventricular ejection fraction (LVEF) and left ventricular fractional shortening (LVFS), *n* (baseline) = 5 (*db/m*), 6 (*db/db*) and 7 (*db/db* + Rol); *n* (4w) = 5 (*db/m*), 5 (*db/db*) and 5 (*db/db* + Rol); *n* (16w) = 5 (*db/m*), 6 (*db/db*) and 5 (*db/db* + Rol); *n* (24w) = 3 (*db/m*), 3 (*db/db*) and 5 (*db/db* + Rol); * *db/db* + Rol *vs db/m*; # *db/db* + Rol *vs db/db*. **d** Levels of cardiac Rol and atRA, *n* = 3, * *vs db/m*. **e** Heart image and heart/tibia ratio of *db/db* mice with Rol, *n* = 5 (*db/m*), 5 (*db/db*) and 4 (*db/db* + Rol), * *vs db/m*. **f** Cardiac Wheat germ agglutinin (WGA), Masson, Terminal deoxynucleotidyl transferase-mediated dUTP nick end labeling (TUNEL), and Transmission electron microscopy (TEM) staining. **g** Analysis of f, *n* (cardiomyocytes area) = 4 (*db/m*), 5 (*db/db*) and 5 (*db/db* + Rol), *n* (collagen area and Tunel⁺ cell) = 3, * *vs db/m*; # *vs db/db*. **h** Body weight, *n* (baseline) = 10 (*db/m*), 10 (*db/db*) and 10 (*db/db* + Rol); *n* (4w) = 10 (*db/m*), 8 (*db/db*) and 9 (*db/db* + Rol); *n* (16w) = 10 (*db/m*), 8 (*db/db*) and 6 (*db/db* + Rol); *n* (24w) = 10 (*db/m*), 4 (*db/db*) and 6 (*db/db* + Rol); * *vs db/m*. **i** Blood glucose, *n* (baseline)=5 (*db/m*), 10 (*db/db*) and 10 (*db/db* + Rol); *n* (4w) = 5 (*db/m*), 6 (*db/db*) and 7 (*db/db* + Rol); *n* (16w) = 5 (*db/m*), 6 (*db/db*) and 5 (*db/db* + Rol); *n* (24w) = 5 (*db/m*), 5 (*db/db*) and 4 (*db/db* + Rol); * vs *db/m*; # *vs db/db*. **j** Serum insulin and homeostasis model assessment of the insulin resistance index (HOMA-IR), *n* = 3 (*db/m*), 5 (*db/db*) and 4 (*db/db* + Rol); * *vs db/m*; # *vs db/db*. **k** Insulin tolerance test (ITT), *n* = 4 (*db/m*), 7 (*db/db*) and 7 (*db/db* + Rol); * *vs db/m*; # *vs db/db*. **l** Serum lipids, *n* (triglyceride, triacylglycerol, triacylglyceride, TG and total cholesterol, TC) = 6 (*db/m*), 5 (*db/db*) and 6 (*db/db* + Rol), n (high density liptein cholesterol, HDL) = 7 (*db/m*), 4 (*db/db*) and 6 (*db/db* + Rol), n (low-density lipoprotein, LDL) = 5 (*db/m*), 4 (*db/db*) and 5 (*db/db* + Rol); * *vs db/m*; # *vs db/db*. Data are expressed as means ± SD. Two-tailed unpaired *t*-test was used for the analysis of statistical significance in d while one-way ANOVA with Tukey post hoc test was used for the analysis of statistical significance in c, e, g h, j, k and l. Source data are provided as a Source Data file. (Black arrows: Typical Tunel stained positive cells; white boxes: lipid droplets).

metabolism disorder (Fig. 5b), prevented heart failure (Fig. 5c and d), attenuated myocardial injury including myocardial hypertrophy, fibrosis, apoptosis, and increases in lipid deposition and mitochondria in the cardiomyocytes (Fig. 5e–g).

These results suggest that RDH10 reduction and cardiac retinol metabolism disorder is the major cause of DCM in T2DM.

## Reduction in RDH10 promoted lipid deposition and free fatty acids (FFAs) uptake mediated by cardiac atRA deficiency in the hearts of T2DM mice

In DM, cardiomyocytes that do not store lipids accumulate large amounts of lipids, which is a process called myocardial steatosis and is a hallmark of cardiac lipotoxicity and DCM[21,22]. Cardiac lipid deposition in DCM is closely associated with excessive cardiac uptake of FFAs[23], which is mainly mediated by Fatty acid translocase (FAT/CD36)[24,25].

In the observation of the ultrastructure of cardiomyocytes, we noticed that altered retinol metabolism status and RDH10 expression were associated with lipid deposition (Fig. 2f, Fig. 3f, Fig. 4f, and Fig. 5f), suggesting a possible link between RDH10, disturbed retinol metabolism and myocardial lipotoxicity in DCM. We performed oil red O staining and cardiac triacylglyceride (TG) levels measurement and found that RDH10 deletion increased myocardial lipid deposition and TG levels whereas AAV9-RDH10 reduced myocardial lipid deposition and TG levels in *db/db* mice (Fig. 6a, b, e, and f), atRA but not Rol reduced myocardial lipid deposition and TG levels in *db/db* mice (Fig. 6c, d, and f), suggesting RDH10 reduction and its leading retinol metabolism disorder promotes lipotoxicity via atRA deficiency in the heart in T2DM. We verified this conclusion again by measuring lipid deposition in Neonatal mouse primary cardiomyocytes (NMPCs), which showed that the silent RDH10-induced increase in lipid deposition could be attenuated by atRA but not Rol when exposed to 200 mmol/L palmitic acid (PA) and that AGN193109 acted as an inhibitor of RARs to prevent the effect of atRA (Fig. 6g). CD36 is a key factor in the abnormal uptake of cardiac FFAs in DCM and has been found to be suppressed by synthetic retinoic acid in atherosclerosis[26]. We found in NMPCs that RDH10 deletion increased both mRNA and protein levels of CD36 by decreasing atRA (Fig. 6h), whereas the increase in cardiac CD36 in *db/db* mice could be restored by supplementing atRA (Fig. 6i), suggesting that in the heart in T2DM, the decrease in RDH10 and its resulting retinol metabolism disorder promotes FFAs uptake. We further performed a cardiac FFAs uptake capacity assay, which showed that cardiac RDH10 deficiency in mice increases cardiac FFAs uptake, while atRA mitigated the increased cardiac FFAs uptake in *db/db* mice (Fig. 6j).

These results suggest that a decrease in cardiac RDH10 increases cardiac lipid deposition and FFAs uptake by reducing cardiac atRA levels, thereby promoting myocardial lipotoxicity in T2DM.

## Ferroptosis is involved in myocardial injury in T2DM

Ferroptosis was identified in 2012 as a form of cell death that relies on iron-catalyzed lipid peroxidation[27]. Recently, Wang Xiang et al. and Ni Tingjuan et al. found that ferroptosis is an important mechanism of diabetic cardiomyopathy in different T2DM mouse models, respectively[28,29]. In our RNA-seq data, we also found that in the hearts of *db/db* mice, ferroptosis-promoting genes HMOX1, MAP2LCB, ACSL1, TRF, CP, NCOA4, SLC39A8, and ALOX15 were gradually upregulated whereas ferroptosis-inhibiting gene SLC7A11 was gradually downregulated (Supplementary Fig. 7a), along with increased levels of 4-Hydroxynonenal (4-HNE), malondialdehyde (MDA) iron, and non-heme iron (indicators of iron atrophy) (Supplementary Fig. 7b-e), further confirmed the presence of ferroptosis in the heart of T2DM mice. In addition, we also confirmed increased iron and 4-HNE levels in the hearts of T2DM patients (Supplementary Fig. 6f and g), demonstrating that ferroptosis also occurs in the hearts of T2DM patients.

## Reduction in RDH10 promoted ferroptosis mediated by suppression of glutathione peroxidase 4 (GPX4), ferroptosis suppressor protein 1 (FSP1) and ferroportin 1 (FPN1) in the hearts of T2DM mice

The study has demonstrated that atRA prevented iron overload-induced liver injury[30], suggesting a link between retinol metabolism and ferroptosis. We measured the levels of cardiac 4-HNE, MDA, iron, and non-heme iron in RDH10-cKO mice and found increased 4-HNE and MDA levels (Fig. 7a–d), which only provided evidence of lipid peroxidation rather than ferroptosis; thus, we further treated RDH10-cKO mice with the ferroptosis inhibitor ferrostatin-1 (Fer-1) to confirm the presence of ferroptosis and found that Fer-1 rescued heart failure and inhibited cardiac 4-HNE accumulation in RDH10-cKO mice (Fig. 7e–h), which suggested that ferroptosis, mainly caused by lipid peroxidation, is involved in myocardial injury in RDH10-cKO mice. To further investigate the molecules that mediate cardiac retinol metabolism disorder leading to ferroptosis, we validated molecules that have been shown to regulate ferroptosis through the regulation of lipid peroxidation, GPX4, FSP1, DHODH, and SLC7A11[31–34]. Our results showed cardiac RDH10 deficiency reduced cardiac GPX4 and FSP1 (Fig. 7i) but not DHODH and SLC7A11 (Supplementary Fig. 7) in RDH10-cKO mice, while in NMPCs, atRA but not Rol reversed silent RDH10-induced GPX4 and FSP1 reduction, which were blocked by AGN193109 (Fig. 7j and k), suggesting that cardiac retinol metabolism disorder-induced atRA deficiency can lead to cardiac ferroptosis by reducing GPX4 and FSP1-mediated increases in lipid peroxidation. Further, we measured the levels of GPX4, FSP1, 4-HNE, and MDA in *db/db* mice with RDH10-AAV9 virus and atRA supplementation and found that reversal of RDH10 and atRA restored GPX4 and FSP1 expression and significantly reduced lipid peroxidation in the hearts of *db/db* mice (Fig. 7l and m), suggesting retinol metabolism

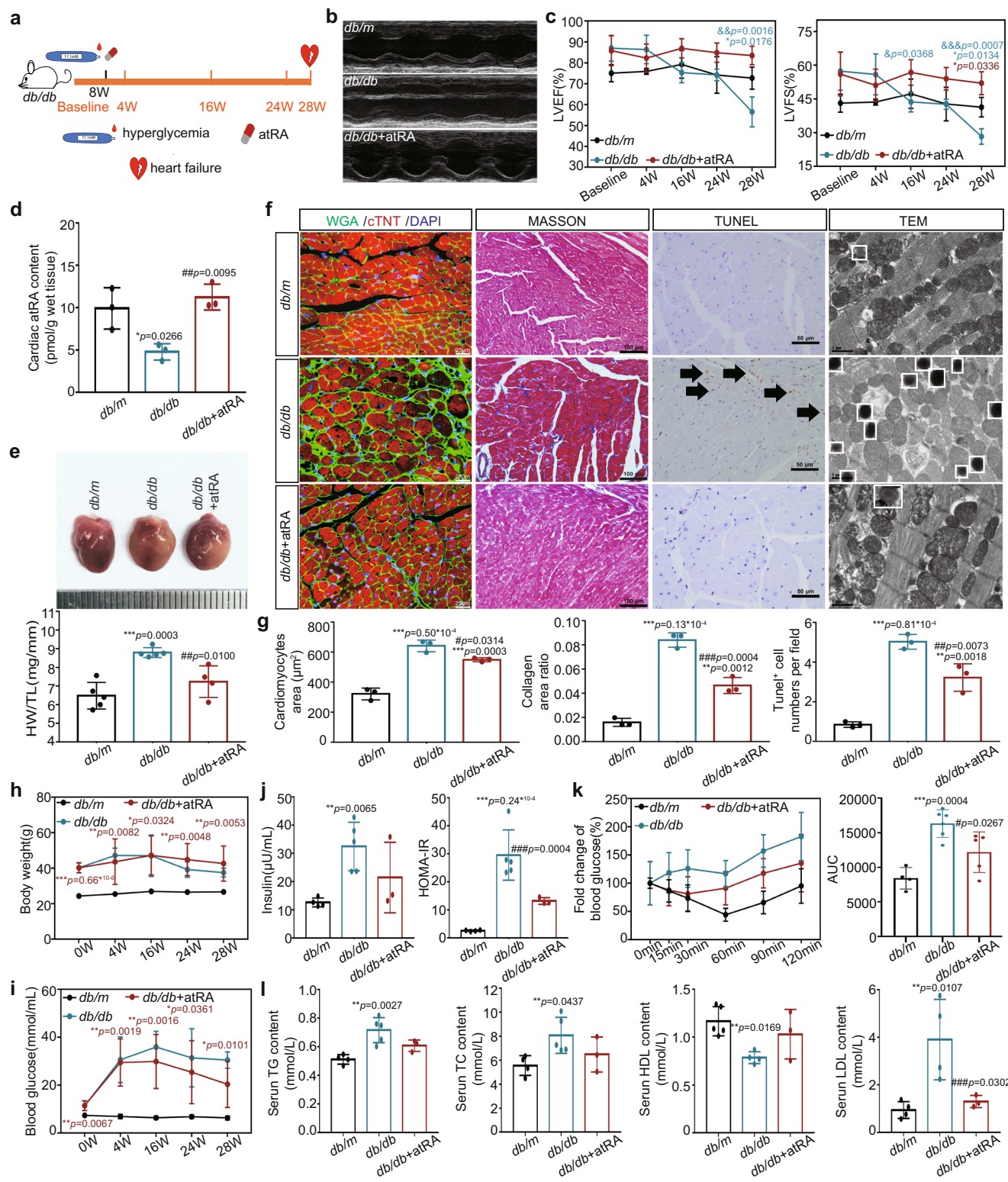

disorder-induced atRA deficiency leads to ferroptosis mediated by GPX4 and FSP1 reduction in the heart in T2DM.

The ferroptosis in the heart in T2DM also exhibited iron accumulation (Supplementary Fig. 6d and e), but we did not observe iron accumulation in the hearts of RDH10-cKO mice (Fig. 7c and d). Wang X, et al. demonstrated that there is an increase in serum iron in T2DM[35], suggesting to us that we should measure iron accumulation in RDH10-cKO mice in the presence of disturbing systemic iron status. We, therefore, fed RDH10-cKO mice a high iron diet (HID) as described[36]

and found that RDH10-cKO mice developed heart failure and cardiac iron accumulation after 3 weeks of HID feeding (Fig. 8a–e). More importantly, AAV-RDH10 and atRA could significantly inhibit cardiac iron accumulation in *db/db* mice (Fig. 8f and g), suggesting that RDH10 deficiency-induced retinol metabolism disorder promotes cardiac iron accumulation via atRA reduction in the heart in T2DM. We measured the levels of transferrin Receptor (TFRC) and FPN1 which regulate iron uptake and output, respectively, and found that alterations in cardiac retinol metabolism altered only the expression of FPN1 (Fig. 8h) but

**Fig. 3 | Cardiac retinol metabolic status, structure, and function as well as serum insulin and lipids in T2DM mice supplemented with atRA (*n* means biologically independent animals).** **a** Schematic diagram of atRA supplementary. **b** Echocardiography. **c** LVEF and LVFS, *n* (baseline) = 3 (*db/m*), 3 (*db/db*) and 5 (*db/db* + atRA); *n* (4w) = 3 (*db/m*), 3 (*db/db*) and 5 (*db/db* + atRA); *n* (16w) = 3 (*db/m*), 4 (*db/db*) and 3 (*db/db* + atRA); *n* (24w) = 5 (*db/m*), 3 (*db/db*) and 3 (*db/db* + atRA); *n* (28w) = 4 (*db/m*), 3 (*db/db*) and 3 (*db/db* + atRA); * *vs db/m*; # *vs db/db*. **d** Levels of cardiac atRA, *n* = 3, * *vs db/m*; # *vs db/db*. **e** Heart image and heart/tibia ratio, *n* = 5 (*db/m* and *db/db*) and 4 (*db/db* + atRA), * *vs db/m*, # *vs db/db*. **f** Cardiac WGA, Masson, Tunel, and TEM staining. **g** Analysis of f, *n* = 3, * *vs db/m*; # *vs db/db*. **h** Body weight, *n* (baseline) = 5 (*db/m*), 5 (*db/db*) and 5 (*db/db* + atRA); *n* (4w) = 5 (*db/m*), 5 (*db/db*) and 5 (*db/db* + atRA); *n* (16w) = 4 (*db/m*), 5 (*db/db*) and 4 (*db/db* + atRA); *n* (24w) = 4 (*db/m*), 4 (*db/db*) and 3 (*db/db* + atRA); *n* (28w) = 4 (*db/m*), 3 (*db/db*) and 3 (*db/db* + atRA); * *vs db/m*. **i** Blood glucose, *n* (baseline) = 5 (*db/m*), 5 (*db/db*) and 5 (*db/db* + atRA); *n* (4w) = 5 (*db/m*), 6 (*db/db*) and 5 (*db/db* + atRA); *n* (16w) = 5 (*db/m*), 4 (*db/db*) and 4 (*db/db* + atRA); *n* (24w) = 5 (*db/m*), 4 (*db/db*) and 5 (*db/db* + atRA); *n* (28w) = 5 (*db/m*), 4 (*db/db*) and 4 (*db/db* + atRA); * *vs db/m*. **j** Serum insulin and HOMA-IR, *n* = 4(*db/m*), 5 (*db/db*) and 3 (*db/db* + atRA), * *vs db/m*; # *vs db/db*. **k** ITT, *n* = 4 (*db/m*), 6 (*db/db*) and 5 (*db/db* + atRA),* *vs db/m*; # *vs db/db*. **l** Serum lipids, *n* (TG and TC) = 4 (*db/m*), 5 (*db/db*) and 3 (*db/db* + atRA), *n* (HDL and LDL) = 5 (*db/m*), 4 (*db/db*) and 3 (*db/db* + atRA), * *vs db/m*, # *vs db/db*. Data are expressed as means ± SD. One-way ANOVA with Tukey post hoc test was used for the analysis of statistical significance. Source data are provided as a Source Data file. (Black arrows: Typical Tunel stained positive cells; white boxes: lipid droplets).

not TFRC (Supplementary Fig. 7), as in the experimental results of other investigators[30].

AtRA function as a ligand for nuclear RARs, RA-RARs can activate or repress transcription of target genes. We predicted that multiple RARs binding sites exist on the promoter sequences of GPX4, FSP1 and FPN1 and verified the effects of altered retinol metabolism on the transcript levels of GPX4, FSP1 and FPN in NMPCs (Supplementary Fig. 8a and b).

These results suggest that a reduction in RDH10 promoted ferroptosis by cardiac atRA-RARs deficiency-induced GPX4, FSP1, and FPN1 reduction in the heart in T2DM.

## Discussion

In this study, we found by RNA-seq analysis that retinol metabolism disorder gradually emerged in the hearts of *db/db* mice with the development of DCM and concluded after a series of experiments that retinol metabolism disorder characterized by Rol overload and atRA deficiency is induced by a decrease in RDH10 and promotes myocardial injury mainly through Rol overload-induced cardiotoxicity and atRA deficiency-induced cardiac lipotoxicity and ferroptosis in the heart in T2DM (Fig. 9a).

We assessed cardiac retinol metabolic status in T2DM mice by measuring the levels of Rol, atRA, and RARs (substrates, products, and major biological effectors of retinol metabolism) and verified retinol metabolism disorder characterized by Rol overload, atRA deficiency, and reduced RARa and RARb in the hearts of T2DM mice. Considering the differences between mice and humans, we also evaluated cardiac retinol metabolic status in T2DM patients. Since human heart tissue is difficult to collect and we could not obtain enough fresh heart tissue to measure the levels of Rol and atRA, we used forensically collected heart sections from T2DM patients to perform immunohistochemistry (IHC) staining of RARs and found that the expression of RARa and RARb were reduced, suggesting that cardiac retinol metabolism disorder may also be present in the hearts of T2DM patients. Additionally, because the samples we used were forensically sourced, we were unable to collect much information about the donors of these samples, which greatly limited our validation in humans. Fortunately, a study by Ni Yang et al. published in early 2021 confirmed the increase in Rol and decrease in atRA in the hearts of patients with heart failure, and this study has filled in the gaps in our validation of humans, although that study focused only on atRA[8].

After demonstrating disturbed cardiac retinol metabolism in T2DM, we assessed the effects of altered Rol and atRA levels on myocardial injury in T2DM by supplementing *db/db* mice with Rol or atRA. Our results showed that Rol, although attenuated systemic metabolic disorder, promoted myocardial injury by exacerbating myocardial fibrosis, apoptosis, and ultrastructural disruption of cardiomyocytes in *db/db* mice, which, combined with the significant upregulation of myocardial Rol levels, suggested that it is cardiotoxicity caused by cardiac Rol accumulation promotes myocardial injury in T2DM. In contrast, atRA attenuated systemic metabolic disorders and myocardial injury and restored cardiac atRA levels in *db/db* mice,

which, combined with a related study on the importance of atRA on the heart[8], suggests that atRA deficiency is a risk factor for myocardial injury in T2DM. Based on these findings, we suggest that cardiac retinol metabolism disorder promotes DCM, atRA is beneficial in the treatment of myocardial injury in T2DM whereas Rol should be avoided in T2DM patients because its cardiotoxicity is overloaded.

As a metabolic substrate in retinol metabolism, Rol has not attracted much attention from researchers in the cardiovascular field, the limited studies available have yielded conflicting conclusions about the roles of Rol in cardiovascular disease, and all of these studies have focused on Rol levels in serum[37]. In this study, we confirmed the beneficial effects of Rol on the systemic metabolic disorder and the cardiotoxicity of Rol overload in the heart.

The retinol metabolic status in the hearts of *db/db* mice indicates the impaired conversion of Rol to atRA. Therefore, we hypothesized and validated that RDH10 deficiency is the initiating factor for cardiac retinol metabolism disorder and myocardial injury in T2DM by overexpressing RDH10 in *db/db* mice and constructing RDH10-cKO mice. RDH10 is a rate-limiting enzyme in retinol metabolism that limits the conversion of Rol to atRA and affects organ development in embryos[11,20], but its function in the adult heart is unknown. In this study, we demonstrated for the first time the roles of RDH10 on cardiac retinol metabolism and heart disease in adulthood. Additionally, the findings associated with RDH10-cKO mice exclude possible effects of abnormalities in blood glucose, lipids, and insulin, indicating the importance of cardiac retinol metabolism in the heart.

Our study verified that cardiac atRA deficiency leads to myocardial injury in T2DM through lipotoxicity and ferroptosis. Lipotoxicity marked by lipid accumulation caused by abnormal FFAs uptake, primarily mediated by CD36, is a key contributor to DCM[21–24]. We verified that in T2DM, alterations in retinol metabolism, particularly atRA levels, regulate cardiac lipid accumulation. In a previous study, researchers suggested that FFAs β-oxidation promoted by atRA/RARs plays the most important role in the regulation of lipid metabolism by retinol metabolism[38]. However, increased FFAs uptake contributes more to cardiac lipotoxicity than other factors in DCM[39]. We considered FFAs uptake as a mechanism by which retinol metabolism affects cardiac lipotoxicity in T2DM and succeeded in preliminarily demonstrating this effect in vitro and in vivo. Based on these results, we suggest that in T2DM, RDH10-mediated retinol metabolism disorder promotes cardiac lipotoxicity by increasing cardiac lipid accumulation and FFAs uptake through reduced atRA.

Ferroptosis is a form of cell death that relies on iron-catalyzed lipid peroxidation and is distinct from necrosis, apoptosis, and autophagy[27]. Relevant studies have shown the involvement of ferroptosis in myocardial injury in T2DM mice[28,29]. We further measured and confirmed the occurrence of ferroptosis in the heart in T2DM mice and patients. Research showed that atRA inhibits iron overload-induced liver injury in mice[30], suggesting a link between retinol metabolism and ferroptosis. We verified that disordered retinol metabolism was associated with cardiac ferroptosis and that atRA supplementation and RDH10 overexpression inhibited cardiac

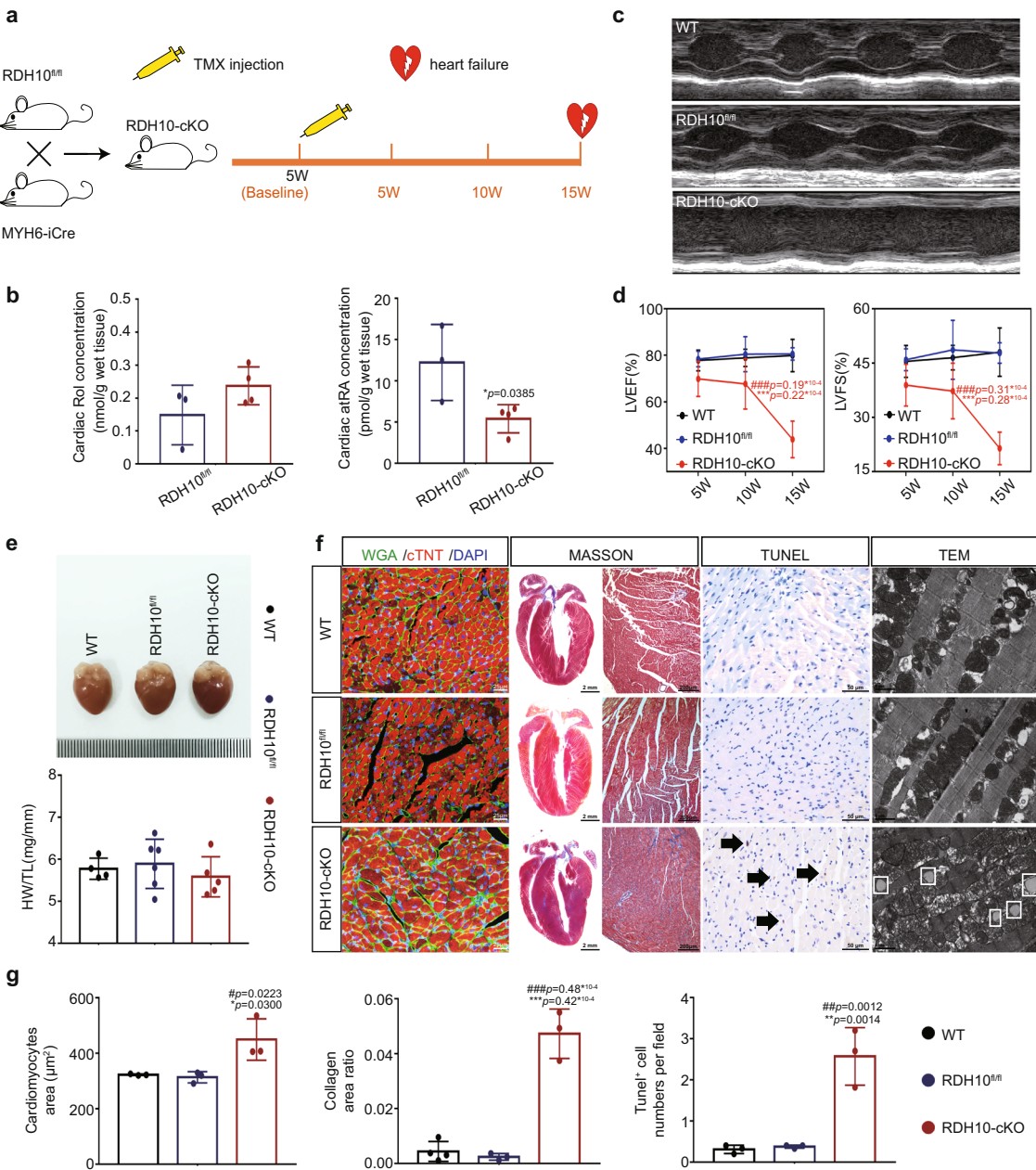

**Fig. 4 | Cardiac retinol metabolic status, structure, and function in cardiomyocyte-specific conditional Retinol Dehydrogenase 10-knockout (RDH10-cKO) mice (n means biologically independent animals). a** Schematic diagram of the construction of RDH10-cKO mice. **b** Cardiac Rol and atRA levels, $n = 3$ (RDH10$^{fl/fl}$) and 4 (RDH10-cKO), * $vs$ RDH10$^{fl/fl}$. **c** Echocardiography. **d** LVEF and LVFS, $n$ (5w) = 5 (WT), 4 (RDH10$^{fl/fl}$), 5 (RDH10-cKO); $n$ (10w) = 5 (WT), 3 (RDH10$^{fl/fl}$), 7 (RDH10-cKO); $n$ (15w) = 4 (WT), 4 (RDH10$^{fl/fl}$), 5 (RDH10-cKO); * $vs$ WT; # $vs$ RDH10$^{fl/fl}$.

**e** Heart image and heart/tibia ratio of RDH10-cKO mice, $n$ = 4 (WT), 6 (RDH10$^{fl/fl}$) and 5 (RDH10-cKO). **f** Cardiac WGA, Masson, Tunel and TEM staining of RDH10-cKO mice. **g** Analysis of f, $n$ = 3, * $vs$ WT; # $vs$ RDH10$^{fl/fl}$. Data are expressed as means ± SD. Two-tailed unpaired $t$-test was used for the analysis of statistical significance in (**b**) while one-way ANOVA with Tukey post hoc test was used for the analysis of statistical significance in (**d**), (**e**) and (**g**). Source data are provided as a Source Data file. (Black arrows: Typical Tunel stained positive cells; white boxes: lipid droplets).

ferroptosis in T2DM mice by reducing iron accumulation and lipid peroxidation. We also found in RDH10-cKO mice that cardiac retinol metabolism disorder mediated by RDH10 deficiency caused ferrptosis by reducing atRA leading to decreases in GPX4 and FSP1 mediating increased lipid peroxidation and decrease in FPN1 mediating cardiac iron accumulation. Based on these results, we conclude that in T2DM, cardiac RDH10 reduction-mediated retinol metabolism disorder leads to ferroptosis, which promotes DCM.

In summary, in this study, we report for the first time that in T2DM, RDH10 reduction leads to cardiac retinol metabolism disorder characterized by Rol overload, atRA deficiency, and RARs reduction, and promotes DCM through Rol overload-induced cardiotoxicity and

atRA deficiency-induced lipotoxicity and ferroptosis, as shown in Fig. 9A. We also suggest that atRA and RDH10 could be potential targets for the prevention and treatment of DCM by correcting disordered retinol metabolism, whereas Rol, as known as vitamin A, should be avoided in patients with T2DM because of its deleterious effects on the heart in excess, as shown in Fig. 9B.

## Methods

### Animal studies

*Db/m*, *db/db* mice (C57BLKS/J background), and MYH6-iCre mice (C57BL/6 background) were purchased from GemPharmatech (Nanjing, Jiangsu, China). RDH10$^{fl/fl}$ mice (C57BL/6 background) were

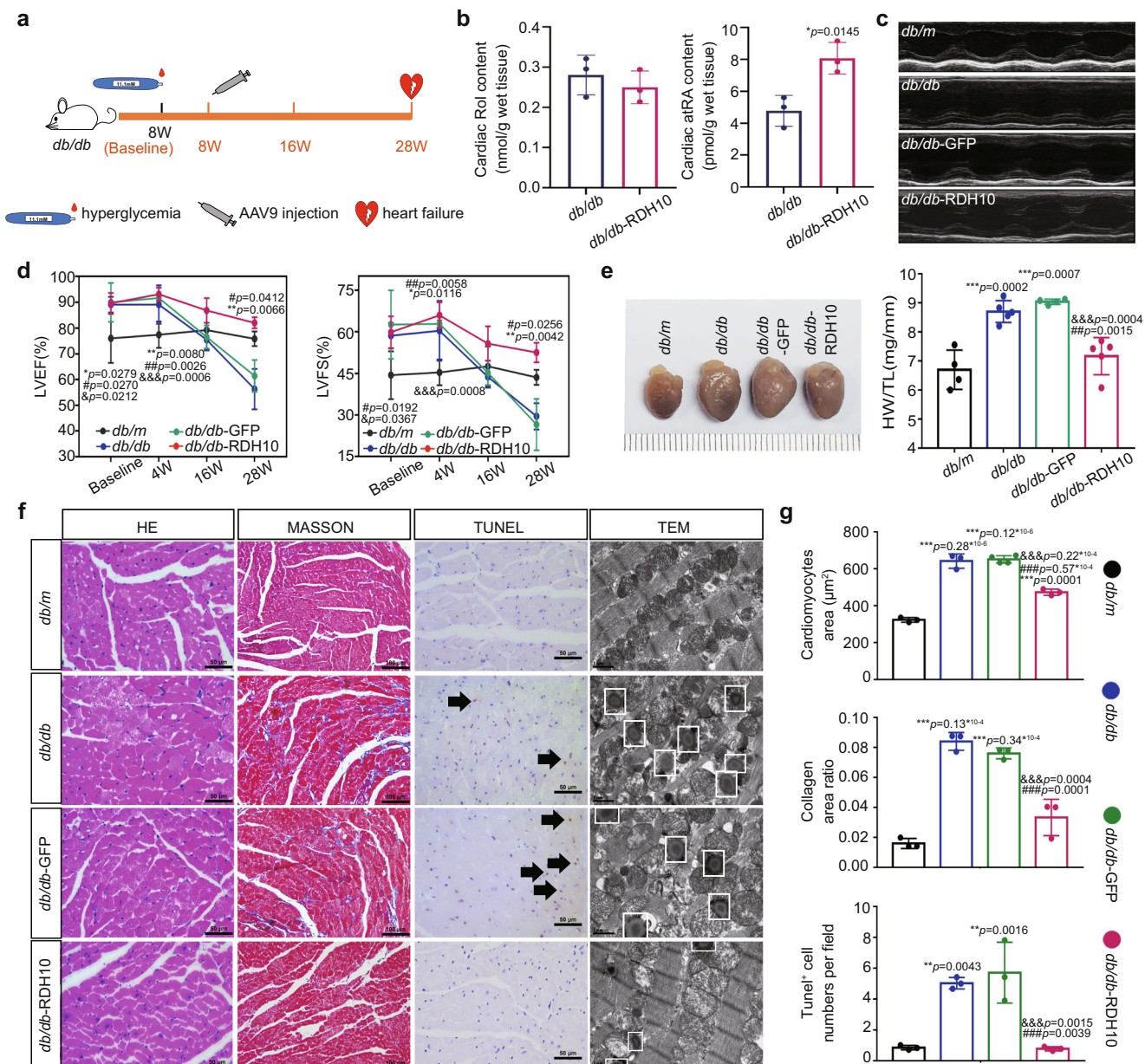

**Fig. 5 | Cardiac retinol metabolic status, structure, and function in T2DM mice injected with via adeno-associated virus 9 (AAV9)-RDH10 (n means biologically independent animals). a** Schematic diagram of AAV9-RDH10 injection. **b** Cardiac Rol and atRA levels, *n* = 3, * *vs db/m*. **c** Echocardiography. **d** LVEF and LVFS, *n* (baseline) = 5 (*db/m*), 5 (*db/db*), 4 (*db/db*-GFP) and 5 (*db/db*-RDH10); *n* (4w) = 5 (*db/m*), 5 (*db/db*), 4 (*db/db*-GFP) and 5 (*db/db*-RDH10); *n* (16w) = 3 (*db/m*), 5 (*db/db*), 4 (*db/db*-GFP) and 3 (*db/db*-RDH10); *n* (28w) = 3 (*db/m*), 3 (*db/db*), 3 (*db/db*-GFP) and 3 (*db/db*-RDH10); * *vs db/db*; # *vs db/db*-GFP; & *vs db/db*-RDH10. **e** Heart image and

heart/tibia ratio, *n* = 4 (*db/m* and *db/db*-GFP) and 5(*db/db* and *db/db*-RDH10), * *vs db/m*; # *vs db/db*; & *vs db/db*-GFP. **f** Cardiac WGA, Masson, Tunel, and TEM staining. **g** Analysis of f, *n* = 3, * *vs db/m*; # *vs db/db*; & *vs db/db*-GFP. Data are expressed as means ± SD. Two-tailed unpaired *t*-test was used for the analysis of statistical significance in (**b**) while one-way ANOVA with Tukey post hoc test was used for the analysis of statistical significance in (**d**), (**e**) and (**g**). Source data are provided as a Source Data file. (White arrows: Typical Tunel stained positive cells; white boxes: lipid droplets).

generously provided by Prof. Jianxing Ma (Health Sciences Center, University of Oklahoma). All enrolled mice were male and aged 6–8 weeks. Mice maintained at the Center for Disease Model Animals of Sun Yat-sen University. Mice were housed on a 12 h light-dark cycle at 22–25 °C with 40–70% humidity and allowed free access to food (chow diet, purchased by Guangdong Medical Laboratory Animal Center, consisting of fat [4.8%], protein [18.6%], and carbohydrate [61%]) and water except as noted. Mice were euthanized by intraperitoneal injection of 150 mg/kg sodium pentobarbital when they reached the experimental time endpoint or when any of the following criteria were met: (1) persistent lethargy and failure to clean hair; (2) failure to respond to physical interventions or behavioral signs of

human touch, including marked inactivity, dyspnea, sunken eyes, and hunched posture; and (3) abnormal central nervous responses (convulsions, tremors, paralysis, head tilt, etc.). Mice were fasted for 12 h before sampling and testing. All animal experiments were approved by the Animal Care and Ethics Committee of Zhongshan School of Medicine, Sun Yat-sen University, and followed the National Institutes of Health Guidelines on the Care and Use of Animals (the protocol number is SYSU-IACUC-2019-B027).

**T2DM mice.** 85 male *db/db* mice were used as T2DM mice and were divided into RNA-seq (15 mice), control (30 mice), Rol treatment (10 mice), atRA treatment (10 mice), AAV9-GFP (10 mice), and AAV9-

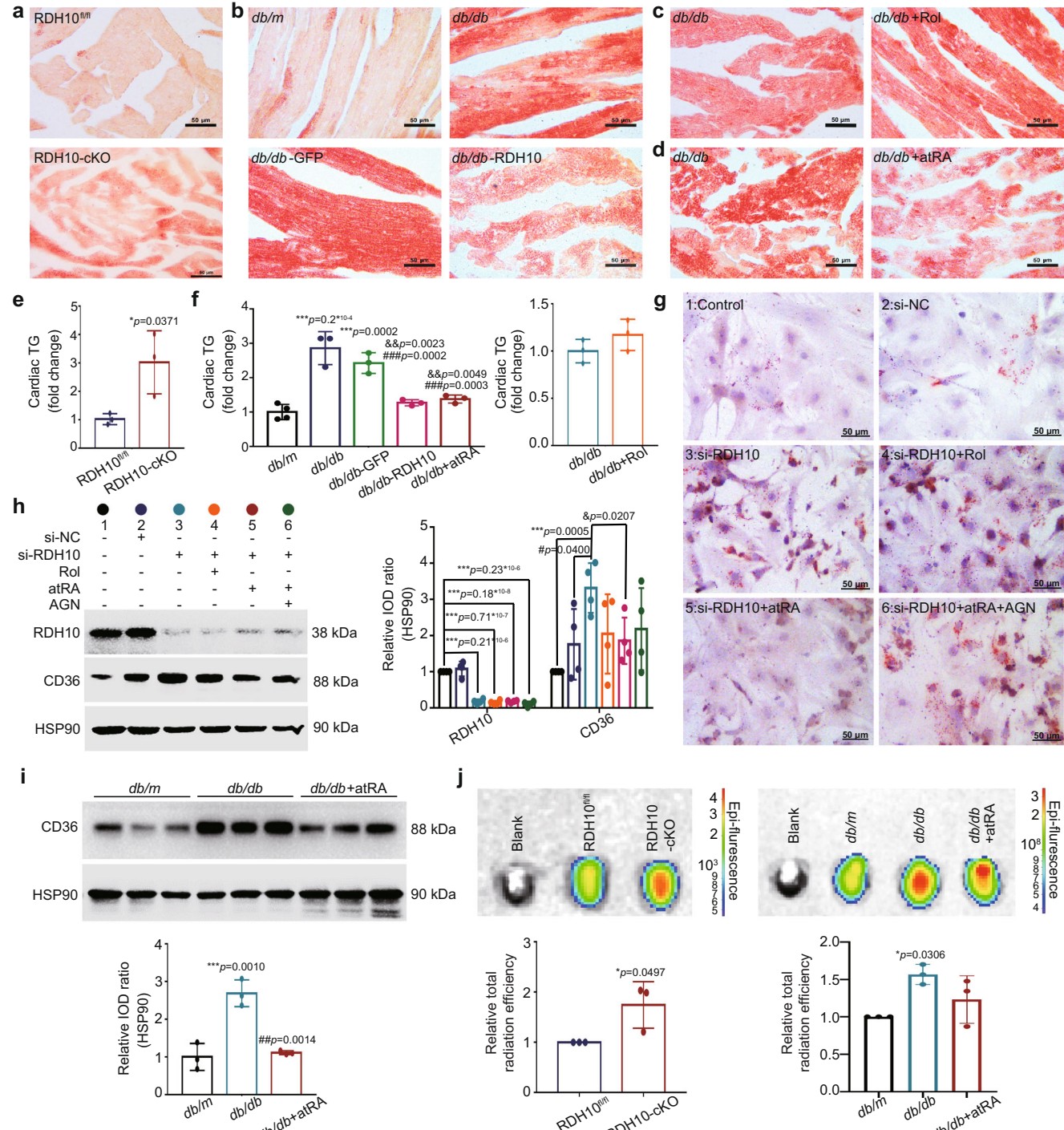

**Fig. 6 | RDH10 reduction-induced retinol metabolism disorder promoted cardiac lipotoxicity via atRA deficiency-mediated increases in lipid accumulation and FFAs uptake in T2DM mice** (*n* means biologically independent animals in e, f, i, and j while n means independent experiments in h). **a** Cardiac oil red O staining of RDH10-cKO mice. **b** Cardiac oil red O staining of *db/db* mice with AAV9-RDH10. **c** Cardiac oil red O staining of *db/db* mice with Rol. **d** Cardiac oil red O staining of *db/db* mice with atRA. **e** Cardiac TG levels of RDH10-cKO mice, *n* = 3, * *vs* RDH10[fl/fl]. **f** Cardiac TG levels of *db/db* mice with AAV9-RDH10, atRA, or Rol, *n* = 4 (*db/m*) and 3 (*db/db, db/db*-GFP, *db/db*-RDH10 and *db/db* + atRA), * *vs db/m*; # *vs db/*

*db*; & *vs db/db*-GFP. **g** Oil red O staining of Neonatal mouse primary cardiomyocytes (NMPCs). **h** WB of CD36 in NMPCs, *n* = 4, * *vs* 1; # *vs* 2; & *vs* 3. **i** WB of cardiac CD36, *n* = 3, * *vs db/m*; # *vs db/db*. **j** Cardiac FFAs uptake capacity detection of RDH10-cKO mice and *db/db* mice with atRA, *n* = 3, * *vs* RDH10[fl/fl] (RDH10-cKO) or *db/m* (*db/db*). Data are expressed as means ± SD. Two-tailed unpaired *t*-test was used for the analysis of statistical significance in (**e**), (**f**) (right) and (**j**) (left) while one-way ANOVA with Tukey post hoc test was used for the analysis of statistical significance in (**f**) (left), (**h**), (**i**) and (**j**) (right). Source data are provided as a Source Data file.

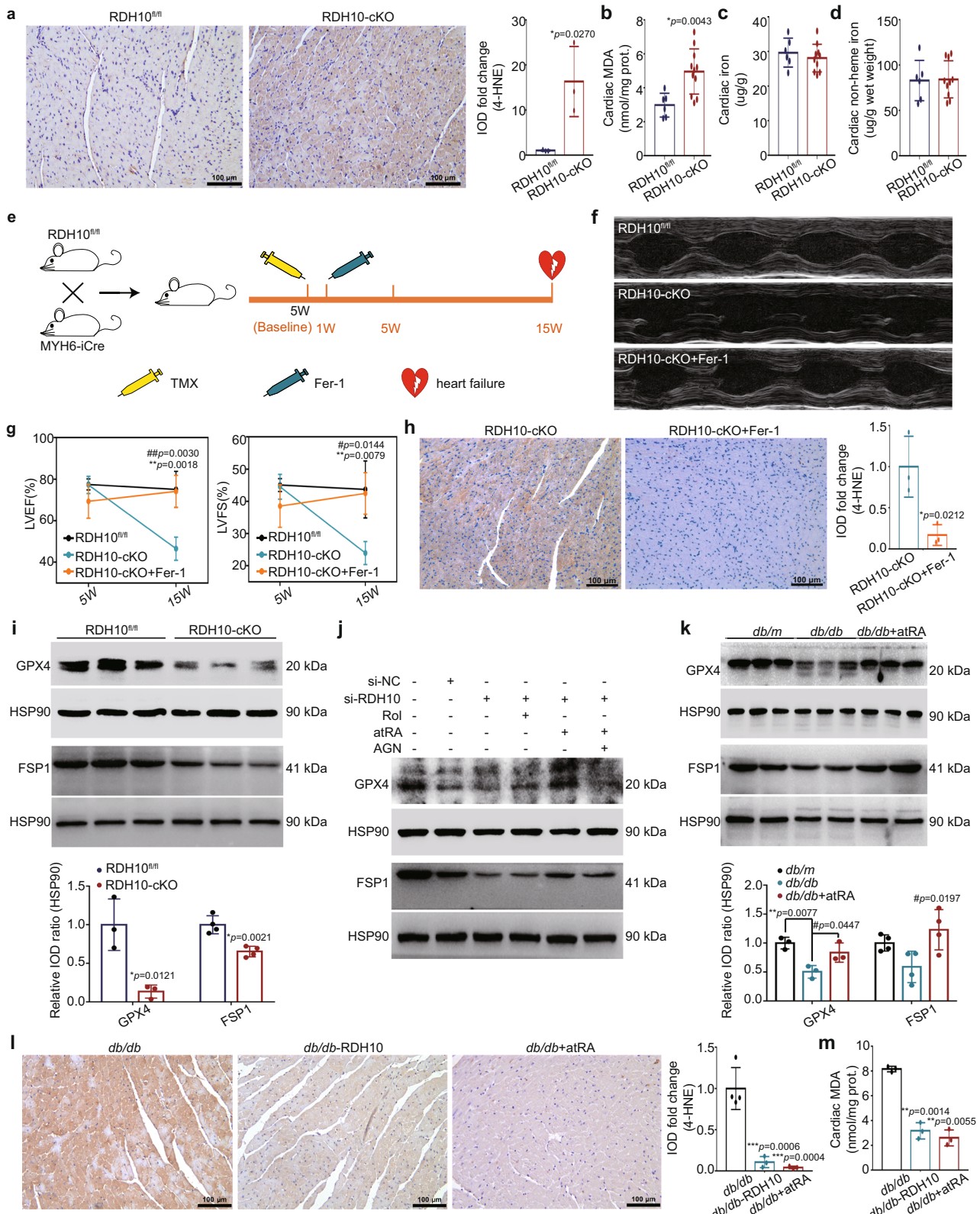

RDH10 (10 mice) groups. 45 age-matched male *db/m* mice were used as normal controls for *db/db* mice.

**RDH10-cKO mice.** RDH10-cKO mice, which contain both RDH10^fl/fl and MYH6-iCre, were bred by RDH10^fl/fl mice and MYH6-iCre mice and were injected tamoxifen intraperitoneally (50 mg/kg, T2859, Sigma -Aldrich, St. Louis, MO) for 5 consecutive days from 5 weeks of age. 55 male

RDH10-cKO mice were divided into control (35 mice), Fer-1 treatment (10 mice), and HID (10 mice) groups. 45 age-matched male RDH10^fl/fl mice that also received TMX injections served as normal controls for RDH10-cKO mice.

**Animal treatments.** Mice in the Rol treatment group received Rol gavage (800 IU/each, 17772, Sigma-Aldrich, St. Louis, MO) every two

**Fig. 7 | RDH10 reduction-induced retinol metabolism disorder promoted ferroptosis via atRA deficiency-mediated glutathione peroxidase 4 (GPX4) reduction in the heart in T2DM (n means biologically independent animals).** **a** Cardiac 4-Hydroxynonenal (4-HNE) staining of RDH10-cKO mice, *n* = 3, * *vs* RDH10$^{fl/fl}$. **b** Cardiac Malondialdehyde (MDA) levels of RDH10-cKO mice, *n* = 6 (RDH10$^{fl/fl}$) and 11 (RDH10-cKO), * *vs* RDH10$^{fl/fl}$. **c** Cardiac iron levels of RDH10-cKO mice, *n* = 6 (RDH10$^{fl/fl}$) and 10 (RDH10-cKO), * *vs* RDH10$^{fl/fl}$. **d** Cardiac non-heme iron levels of RDH10-cKO mice, *n* = 6 (RDH10$^{fl/fl}$) and 10 (RDH10-cKO). **e** Schematic diagram of Fer-1 treatment in RDH10-cKO mice. **f** Echocardiography. **g** LVEF and LVFS, *n* (5w) = 3 (RDH10$^{fl/fl}$), 4 (RDH10-cKO), 5 (RDH10-cKO+Fer-1);* *vs* RDH10$^{fl/fl}$; # *vs*

RDH10-cKO+Fer-1. **h** Cardiac 4-HNE staining of RDH10-cKO mice with Fer-1, *n* = 3, * *vs* RDH10-cKO. **i** WB of cardiac GPX4 and FSP1 in RDH10-cKO mice, *n* = 3, * *vs* RDH10$^{fl/fl}$. **j** WB of GPX4 amd FSP1 in NMPCs. **k** WB of cardiac GPX4 and FSP1 in *db/db* mice with atRA, *n* = 3, * *vs db/m*; # *vs db/db*. **l** Cardiac 4-HNE staining of *db/db* mice with AAV9-RDH10 or atRA, *n* = 4 (*db/db*), 3 (*db/db* + RDH10) and 3 (*db/db* + atRA), * vs *db/db*. **m** Cardiac MDA levels of *db/db* mice with AAV9-RDH10 or atRA, *n* = 3, * vs *db/db*. Data are expressed as means ± SD. Two-tailed unpaired *t*-test was used for the analysis of statistical significance in (**a**), (**b**), (**c**), (**d**), (**h**) and (**i**) while one-way ANOVA with Tukey post hoc test was used for the analysis of statistical significance in (**g**), (**k**) and (**l**). Source data are provided as a Source Data file.

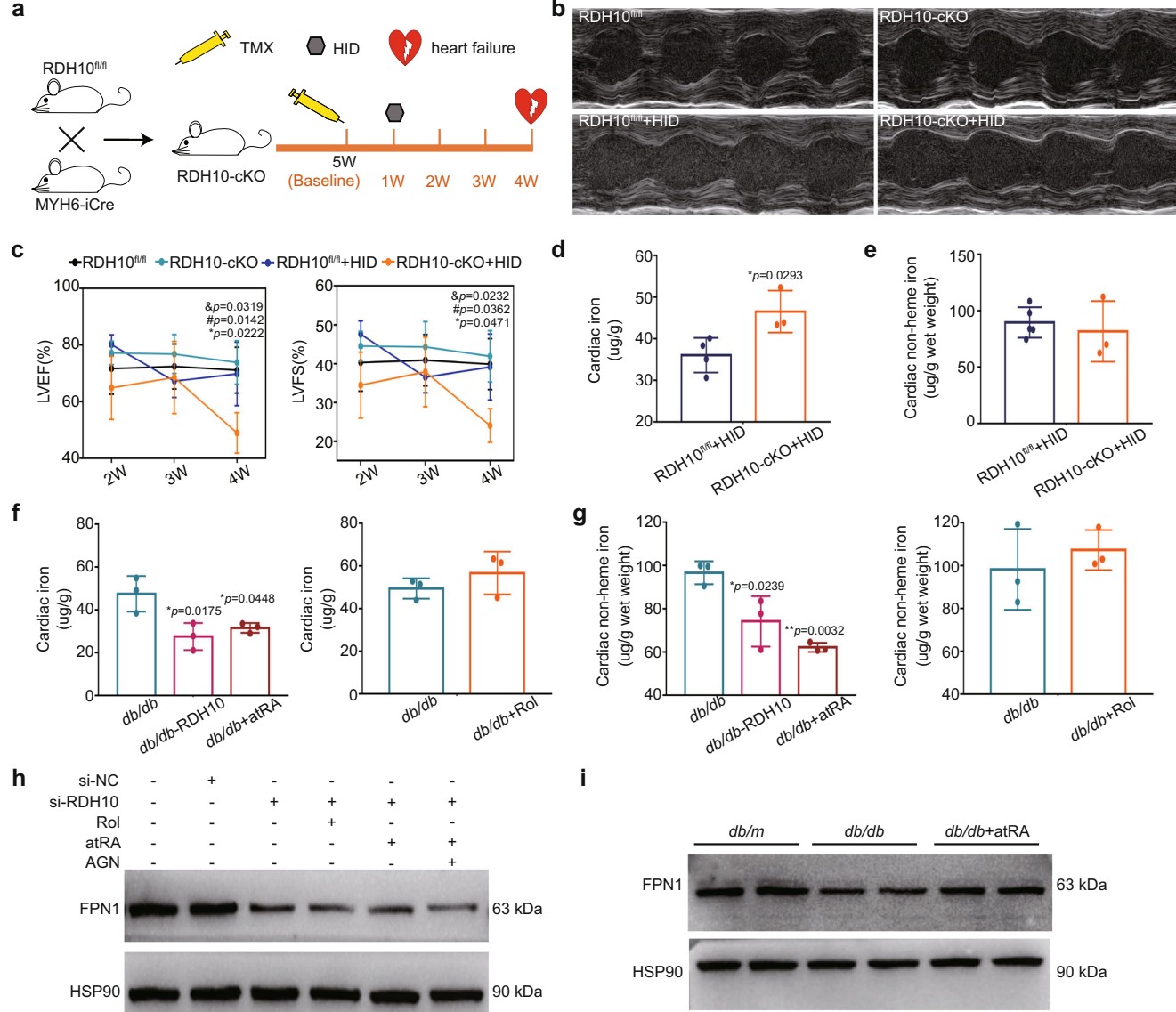

**Fig. 8 | RDH10 reduction-induced retinol metabolism disorder promoted iron accumulation via atRA deficiency in the heart in T2DM (*n* means biologically independent animals).** **a** Schematic diagram of high iron diet (HID) treatment in RDH10-cKO mice. **b** Echocardiography. **c** LVEF and LVFS, *n* (2w) = 3 (RDH10$^{fl/fl}$), 4 (RDH10-cKO), 3 (RDH10$^{fl/fl}$ + HID) and 3 (RDH10-cKO+HID); *n* (3w) = 3 (RDH10$^{fl/fl}$), 3 (RDH10-cKO), 4 (RDH10$^{fl/fl}$ + HID) and 3 (RDH10-cKO+HID); *n* (4w) = 5 (RDH10$^{fl/fl}$), 4 (RDH10-cKO), 5 (RDH10$^{fl/fl}$ + HID) and 3 (RDH10-cKO+HID); * *vs* RDH10$^{fl/fl}$; # *vs* RDH10-cKO; & *vs* RDH10$^{fl/fl}$ + HID. **d** Cardiac iron levels, *n* = 4 (RDH10$^{fl/fl}$) and 3 (RDH10-cKO), * *vs* RDH10$^{fl/fl}$ + HID. **e** Cardiac non-heme iron levels, *n* = 55 (RDH10$^{fl/fl}$)

and 3 (RDH10-cKO). **f** Cardiac iron levels of *db/db* mice with AAV9-RDH10, Rol or atRA, *n* = 3, * *vs db/db*. **g** Cardiac non-heme iron levels of *db/db* mice with AAV9-RDH10, Rol or atRA, *n* = 3, * *vs db/db*. **h** WB of FPN1 in NMPCs, these results were independently repeated 3 times with similar results. **i** WB of cardiac FPN1 in *db/db* mice with atRA, these results were independently repeated 3 times with similar results. Data are expressed as means ± SD. Two-tailed unpaired *t*-test was used for the analysis of statistical significance in (**d**), (**e**), (**f**) (right) and (**g**) (right) while one-way ANOVA with Tukey post hoc test was used for the analysis of statistical significance in (**c**), (**f**) (left) and (**g**) (left). Source data are provided as a Source Data file.

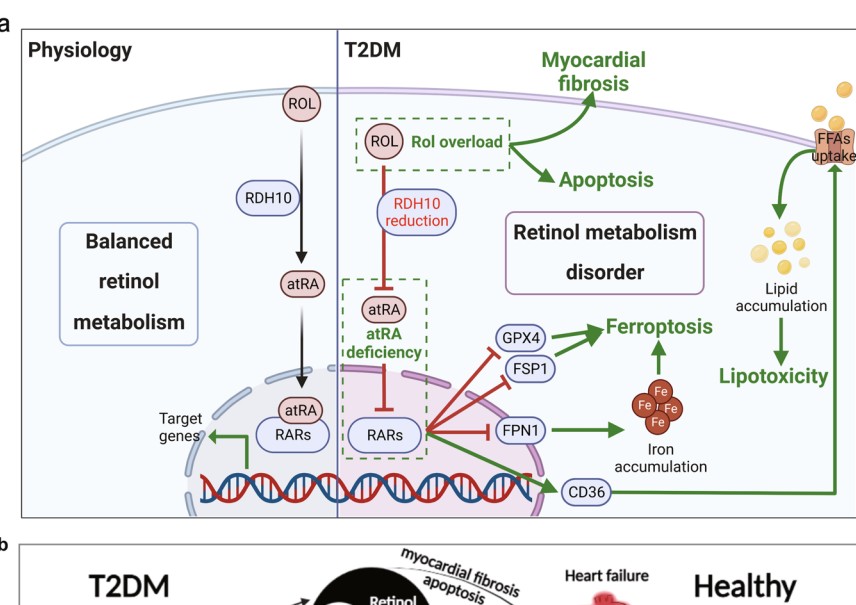

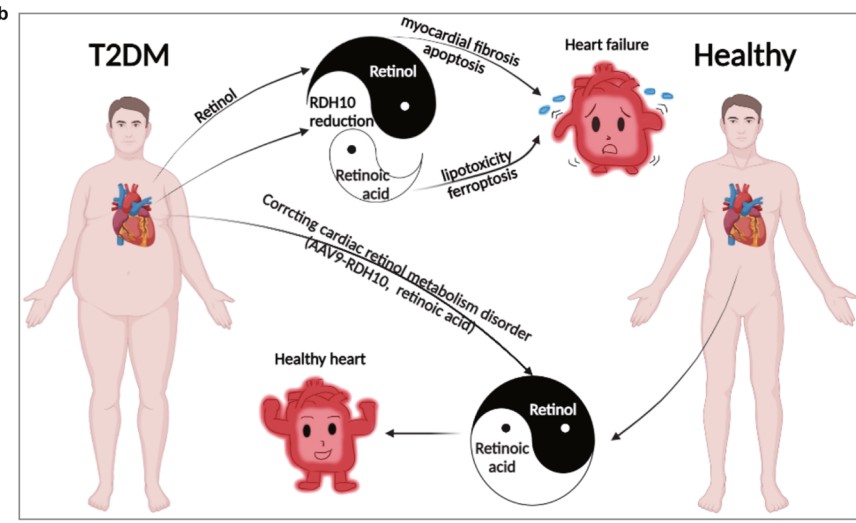

**Fig. 9 | Graphical abstract. a** Graphical summary of the mechanistic study of this study. **b** Graphical summary of clinical implications of this study. Graphics were created with Biorender.com.

days from 8 weeks of age. Mice in the atRA treatment group received atRA intraperitoneal injection (5 mg/kg body weight, R2625, Sigma-Aldrich, St. Louis, MO) daily from 8 weeks of age. Mice in the Fer-1 treatment group received Fer-1 intraperitoneal injection (SML0583, 1 mg/kg body weight, Sigma-Aldrich, St. Louis, MO) daily from 6 weeks of age. Mice in HID group were fed HID (8.3 g carbonyl iron/kg, RD17082801, ReadyDietech, Shenzhen, China) from 6 weeks of age.

**Drugs preparation.** Rol, atRA, and Fer-1 were accurately weighed and placed in brown light-proof tubes and dissolved with dimethylsulfoxide (DMSO) (ST2335, Beyotime, Shanghai, China) followed by diluting 100-fold with corn oil (C116023, Aladdin, Shanghai, China) to prepare working solutions with the final concentrations of 5 mg/mL, 2.5 mg/mL and 0.4 mg/mL, respectively.

**Gene therapy.** A recombinant AAV9 vector carrying the mouse RDH10 sequence (AAV9-RDH10, DZ-AAV-Rdh10-OE, Dongze, Hanbio Inc, Shanghai, China) was used to overexpress RDH10. AAV9-GFP (DZ-AAV-Rdh10-NC, Dongze, Hanbio Inc, Shanghai, China) was used as a negative control. 0.8*10^11 vg/per animal of AA9-RDH10 or AAV9-GFP was transferred into T2DM mice, respectively, by tail vein injection at the age of 16 weeks.

## Echocardiography
Mice were anesthetized with 1.5% isoflurane and placed on a thermostat at 37 °C immediately, 0.5% isoflurane is inhaled continuously to prevent them from waking up. For image acquisition, Vevo 3100 Imaging System (VisualSonics, Canada) with a 400 MHz probe was used to detect cardiac motion in the long-axis view, then the probe was rotated 90 degrees to detect cardiac motion in the short-axis view, and graphs were acquired in M-Mode near the papillary muscles. The heart rates of the mice were controlled at 450–600 bpm.

## RNA-seq
Total RNA was extracted from the hearts of *db/m* mice and *db/db* mice at the age of 4, 24, and 32 weeks using the RNeasy Mini Kit (74104, Qiagen, Duesseldorf, Germany). Then, the strand-specific library was prepared after rRNA depleted and RNA fragmentation, First Strand cDNA synthesizing, Second Strand cDNA synthesizing, 3' ends Adenylating, adapter Ligation, and PCR amplification. RNA and the library preparation integrity were inspected with Agilent Bioanalyzer 2100 (Agilent Technologies, Santa Clara, CA, USA). And the cluster and first dimension sequencing primer hybridization were accomplished on the cBot of the Illumina sequencing machine. Finally, performed sequencing at SHBIO Corporation.

The analysis of the KEGG pathway and GO terms were completed using KOBAS (http://kobas.cbi.pku.edu.cn/) online[40].

## Cardiac Rol, atRA, and retinyl ester content measurements

Cardiac Rol, atRA, and retinyl ester measurements were performed as described by others[41], and the details of the experiment were as follows.

**Sample preparation.** Homogenized the heart tissue in 200 μL of cold NaCl solution (0.9%) for 5 s, add 100 μL retinol-D4 (IR-23012, IsoSciences, Ambler, PA, USA) as internal standard for Rol and atRA, and retinyl acetate (46958, Sigma-Aldrich, St. Louis, MO) as internal standard for retinyl esters, then homogenize for another 5 seconds. For Rol and at RA, 1 mL formic acid n-hexane solution (1%) was added to the homogenizing to facilitate phase separation by a 5 min vortex and 5 min centrifugation (16,200 g). The supernatants were collected in a new tube and blown dry with nitrogen followed by re-dilution with 100 μL 70% methanol. For retinyl esters, 24 μL 0.5 M NaOH solution was added to tissue homogenate, followed by 1 mL n-hexane. The mixture was votexed 5 min and centrifuged at 16,200 g for 5 min. The supernatants were collected in a new tube and blown dry with nitrogen followed by re-dilution with 100 μL 100% methanol. The processes above should be protected from the light.

**LC-MS/MS analysis.** Samples were carried out on a Sciex Jasper TM high performance liquid chromatography (HPLC) system (Sciex, M.A., U.S.A) which consisted of a solvent degasser, a binary pump, an autosampler, and a column oven. The LC system was coupled with a Sciex Triple Quad TM 4500MD MS (Sciex, M.A., U.S.A) in ESI ionization mode. Data were acquired and analyzed with Analyst® MD version 1.6.3 and Multi Quant TM MD version 3.0.2 (Sciex, M.A., U.S.A.). For Rol and at RA, LC was performed on a Phenomenex Kinetex C18 (50 * 2.1 mm, 2.6 μm, Phenomenex, CA, USA). Mobile phase A consisted of acetonitrile/methanol/water (4:3:3, v/v/v) with 0.1 % formic acid, and mobile phase B containing acetonitrile/methanol/water (5.5:3:1.5, v/v/v) with 0.1% formic acid at a column oven temperature of 25 °C. The flow rate was set to 0.2 ml/min and the gradient elution procedure was as follows: initial conditions 75% B; from 0 to 3.5 min linear increase to 95% B; between 3.5 and 4.5 min 95% B was retained; at 4.6 min back to initial conditions with 75% B; finally, 75% B was held from 4.6 to 5.9 min. For retinyl esters, LC was performed on a Phenomenex Kinetex C18 (100 * 2.1 mm, 1.7 μm, Phenomenex, CA, USA). Mobile phase A consisted of acetonitrile containing 0.1% formic acid and 5 mM ammonium acetate, and mobile phase B containing pure water with 0.1% formic acid and 5 mM ammonium acetate at a column oven temperature of 40 °C. The flow rate was set to 0.3 ml/min with the gradient elution as follows: initial conditions 85% B for 1.5 min, from 0 to 9 min linearly increased to 100% B, and then decreased to 85% at 10 min, held for another 1 min. The autosampler was set at 10 °C. The injection volume was 10 μL.

**Mass spectrometry.** The MS conditions were as follows: electrospray ionization (ESI) under positive mode; nebulizer gas: nitrogen; curtain gas, 30 psi; ion spray voltage, 4000 V; temperature, 400 °C; gas 1 and gas 2, 35 and 40 psi, respectively; collision gas, 10 psi. The parameters of the mass spectrometer were optimized, and the multiple reaction monitoring (MRM) transitions of retinol, all-trans-retinoic acid, as well as D4-retinol were chosen as 269.2 > 93.1, 273.1 > 94.0, and 301.2 > 123.1, respectively. The MRM transitions for retinyl esters were chosen as 329.3 > 269.3 for retinyl acetate, 524.4 > 268.1 for retinyl palmitate (16:0), 552.5 > 268.2 for retinyl stearate (18:0), 522.4 > 268, retinyl palmitoleate (16:1), and retinyl oleate (18:1). The quantification was performed using the calibration curve.

## RDH activity assay

Cardiac RDHs activity assay was performed as described by others[41], and the details of the experiment were as follows.

100 μg of cardiac microsomal fractions protein was incubated with 3 μM all-trans-retinol (17772, Sigma-Aldrich, St. Louis, MO) solubilized with bovine serum albumin and 1 mM NAD + (NAD98-RO, Sigma-Aldrich, St. Louis, MO) in 0.5 ml of the reaction buffer for 20 min at 37 °C. Reactions were stopped by the addition of an equal volume of ice-cold methanol, and Retinaldehyde were extracted twice with 2 ml of hexane. Hexane layers were dried, and the dry residue was reconstituted in 0.2 ml of acetonitrile. Retinaldehyde were separated by normal-phase HPLC using Spherisorb S3W column (4.6 mm × 100 mm; Waters) and isocratic mobile phase consisting of acetonitrile at 1 ml/min and analyzed.

## Western blotting analysis

The heart tissues or neonatal mouse primary cardiomyocytes (NMPCs) were lysed with RIPA buffer (P0013B, Beyotime, Shanghai, China) for total protein extraction. The protein concentration was determined using the BCA protein assay kit (71285, Millipore, Bedford, MA, USA) according to the manufacturer's protocol. 35 μg protein was subjected to SDS-PAGE for electrophoresis, transferred to 0.45 μm PVDF membranes (IPVH00010, Millipore, Bedford, MA, USA), and immunoblotted with antibodies. The bands were quantified using the ImageJ software program (v.1.45, National Institutes of Health, Bethesda, MD, USA). The antibodies are listed in Supplementary Table 1.

## Serum measurement

Serum was obtained by centrifugation (4 °C, 1,500 g, 20 min) of blood collected from the eye sockets of the mice and stored at −80 °C. The serum fasting insulin (FINS) levels were examined using an enzyme-linked immunosorbent assay (CSB-E05071m, Cusabio, Wuhan, Hubei, China). The serum TC, TG, LDL-c, and HDL-c levels were examined using commercial reagent kits (A111-1-1, A110-1-1, A113-1-1 and A1122-1-1, Jiancheng, Nanjing, Jiangsu, China).

## Homeostasis model assessment of the insulin resistance index (HOMA-IR) and Insulin tolerance test (ITT)

The fasting blood glucose (FBG) levels were examined after the mice fasted for 12–16 h, and the HOMA-IR was calculated with the equation (FBG (mM/L) × FINS (mIU/L)) / 22.5. To perform the ITT, 1 U/kg body weight of insulin (Novolin R, Novo Nordisk, Bagsvaerd, Denmark) was intraperitoneally injected into the mice, then the blood glucose levels were examined at 0, 15, 30, 60, 90, and 120 min after the injection. The change curve of the blood glucose level was drawn, and the AUC was calculated.

## Human samples

Heart samples from 11 healthy populations and 11 age-matched T2DM patients (age difference ≤5) used in this study were collected from the National Center for Medico-legal Expertise of Sun Yat-sen University. All sample donors were diagnosed without coronary heart disease and hypertension. The use of human heart samples was approved by the ethics committee of Zhongshan School of Medicine, Sun Yat-sen University with the protocol number (2019-B027) and all data and sample collection were in strict accordance with ethics guidelines of Zhongshan School of Medicine, Sun Yat-sen University. Informed consent was obtained from the legal representatives of the victims. The principles outlined in the Declaration of Helsinki were followed. The information details of the donors were provided in Supplementary Table 3.

## Immunohistochemistry (IHC)

Heart samples were collected and fixed overnight in 4% paraformaldehyde (BL539A, Biosharp, Hefei, Anhui, China), followed by

routine dehydration and sectioning (5 µm). Sections were blocked with 3% hydrogen peroxide, antigen repaired with citrate buffer (P0083, Beyotime, Shanghai, China), permeabilized with 0.3% Triton-100 (ST795, Beyotime, Shanghai, China), blocked with 5% BSA (A1933, Sigma-Aldrich, St. Louis, MO), incubated with primary antibody overnight at 4 °C and incubated with HRP-labeled secondary antibody for 30 min at 37 °C. Visualization was performed under the microscope (DFC700T, Leica, Germany) with DAB Horseradish Peroxidase Color Development Kit (P0203, Beyotime, Shanghai, China). Finally, the sections were sealed with neutral balsam fixative (G8590, Solarbio, Beijing, China). The positive cell number was counted from 4–5 fields per sample with the ImageJ software program (v.1.45, National Institutes of Health, Bethesda, MD, USA) and the mean density was quantified from 4–5 fields per sample with the Image-pro plus software program (v.6.0, Media Cybernetics, Rockville, MD, USA). The antibodies are listed in Supplementary Table 1.

### Immunofluorescence
Heart samples were collected and fixed in 4% paraformaldehyde (BL539A, Biosharp, Hefei, China) overnight followed by conventional dehydration and slicing (5 µm). The heart samples sections were blocked with 3% hydrogen peroxide and then performed at 95 °C for 10 min using citrate buffer (P0083, Beyotime, Shanghai, China), 0.3% Triton-100 (ST795, Beyotime, Shanghai, China) was used for permeabilization and then the blocking step was carried out using the 5% BSA (A1933, Sigma-Aldrich, St. Louis, MO). After overnight incubation of the primary antibody at 4 °C, a secondary antibody was applied to the sections at 37 °C for 90 min. All the immune-fluorescence images were captured by a fluorescence microscope (DFC700T, Leica, Germany). The antibody is listed in Supplementary Table 1.

### WGA-FITC staining
The cTNT was staining first following the method of Immunofluorescence, and then the sections were incubated with WGA-FITC (W11261, Sigma-Aldrich, St. Louis, MO) at 37 °C for 30 min. All the immune-fluorescence images were captured by a fluorescence microscope (DFC700T, Leica, Germany). The cardiomyocyte area was calculated from 4–5 fields per sample with the ImageJ software program (v.1.45, National Institutes of Health, Bethesda, MD, USA).

### Masson staining
Standard Masson staining was performed using the Modified Masson's Trichrome Stain Kit (G1345, Solarbio, Beijing, China) according to the manufacturer's protocol. The positive area was quantified from several fields per sample with the Image-pro plus software program (v.6.0, Media Cybernetics, Rockville, MD, USA).

### Terminal deoxynucleotidyl transferase-mediated dUTP nick end labeling (TUNEL) staining
TUNEL staining was performed using the Colorimetric TUNEL Apoptosis Assay Kit (C1091, Beyotime, Shanghai, China). The positive cell numbers were counted from 4–5 fields per sample with the ImageJ software program (v.1.45, National Institutes of Health, Bethesda, MD, USA).

### Transmission electron microscopy (TEM)
Cardiac ultrastructure was examined under a transmission electron microscope (Tecnai G2 Spirit Twin +GATAN 832.10 W; FEI; Czech Republic) using conventional methods. In brief, heart tissues were fixed with 2.5% glutaraldehyde in 0.1 mol/L phosphate buffer (pH 7.4), followed by 1% $OsO_4$. After dehydration, thin sections were stained with uranyl acetate and lead citrate for observation, images were acquired digitally.

### Oil red O staining
The heart tissues were fixed in 4% paraformaldehyde (BL539A, Biosharp, Hefei, China) overnight then dehydrated with a sucrose gradient, and embedded in the Tissue-Tek OCT compound (4583, Sakura Finetek, Tokyo, Japan). Then, the sections (8 µm) were stained with oil red O (O0625, Sigma-Aldrich, St. Louis, MO, USA) for 15 min.

### Tissues FFAs uptake fluorescence imaging
As described by others[42], mice were injected with 1 µg/g body weight BODIPY™ 558/568 C12 (D3835, Thermo Fisher, MA, USA) via the tail vein, 50 min later, the hearts were collected after removing fat, blood, and auricles and rinsed in PBS. X-ray and fluorescence imaging was performed using a small animal living fluorescence imaging system (IVIS Spectrum, PerkinElmer, USA).

### Cardiac MDA measurement
Cardiac MDA measurement was performed using the Lipid Peroxidation MDA Assay Kit (S0131, Beyotime, Shanghai, China) according to the manufacturer's protocol.

### Cardiac iron measurement
Cardiac iron measurement was performed using the Tissue Iron Content Assay Kit (BC4355, Solarbio, Beijing, China) according to the manufacturer's protocol.

### Cardiac non-home iron measurement
Cardiac Non-home iron measurement was performed as described by others[43] as follows. Weighed and digested the heart tissues in NHI acid (10% trichloroacetic acid in 3 M HCl) for 48 h at 65–70 °C. An equal volume of samples, iron standard (500 µg/dl, Aladdin, Shanghai, China) or NHI acid were incubated with 200 µl BAT buffer (0.2% thioglycolic acid and 0.02% disodium 4,7 diphenyl 1,10 phenanthroline disulfonate in 50% saturated NaAc solution) for 10 min at room temperature. The absorbance of the mixtures was read at 535 nm and the absorbance of the standard was used to scale the unknown sample concentration.

### Perl's Prussian blue staining
Perl's Prussian blue staining was performed using the Prussian Blue Iron Stain Kit (with Eosin solution) (G1424, Solarbio, Beijing, China) according to the manufacturer's protocol.

### Quantitative real-time polymerase chain reaction (Q-PCR)
Total RNA was isolated using the RNAiso Plus (9109, Takara, Tokyo, Japan). 0.5 mg of total RNA was reverse transcribed into complementary DNA using the HiScript III RT SuperMix for qPCR (+gDNA wiper) kit (R323, Vazyme, Nanjing, China). Then the Q-PCR was performed on the ABI Q6 Flex Real-Time PCR machine (Applied Biosystems, Foster City, CA, USA) using the 2x SYBR Green qPCR Master Mix kit (B21203, Bimake, Houston, Texas USA). The relative gene expression levels were analyzed using the 2(−ΔΔCt) method and normalized against 18 S expression. Primer sequences are in Supplementary Table 2.

### Neonatal mouse primary cardiomyocytes (NMPCs) isolation and culture
1–3-day-old neonatal mice were skin disinfected with 75% ethanol, then removed their hearts by cutting with sharp forceps and quickly minced in ice-cold D-hanks. The shredded tissue was digested with 0.1% trypsin for 6–10 h at 4 °C, followed by 3–5 digestions with 0.08% collagenase type II (17101015, Gibco, Grand Island, NY, USA) for 10 min each (37 °C, 80 rpm). The precipitate was transferred to a new tube and neutralized with a volume of DMEM (C11995500BT, Gibco, Grand Island, NY, USA) containing 10% FBS (Gibco, Grand Island, NY, USA) and the digestion was continued by adding new type II collagenase until no precipitate

was evident. The collected supernatant was centrifuged (160 g, 5 min), and the precipitate was resuspended in D-hanks and filtered through a 70 μm cell strainer (352350, Falcon, Corning, NY, USA). The filtered cytosol was centrifuged again (160 g, 5 min) and the precipitate was resuspended in DMEM containing 10%FBS. The resuspended cells were grown in 15 cm$^2$ cell culture plates for 1.5 h to remove non-cardiomyocytes. Finally, purified cardiomyocytes were cultured in DMEM containing 10% FBS and BrdU (19–160, Sigma-Aldrich, St. Louis, MO, USA) in cell culture dishes or Millicell EZ SLIDE 8 well glass slides (PEZGS0816, Merck Millipore, Billerica, MA, USA) coated with attachment factor (S006100, ThermoFisher, Waltham, MA, USA), and maintained in an incubator at 37 °C and 5% CO$^2$.

### In vitro treatments

Cells were cultured till the morphological expansion and autonomic rhythmic contractions were observed, then changed the culture medium to serum-free medium 12 h before the subsequent experimental treatments.

**Small interfering RNA (si-RNA) transfection.** Mouse RDH10 stealth siRNA (1320001, Invitrogen, Carlsbad, CA, USA) was transfected with NMPCs using the Lipo3000 Transfection Kit (L3000015, Invitrogen, Carlsbad, CA, USA) according to the manufacturer's protocol.

**Drugs treatment.** After a 36 h si-RNA transfection, cells were cultures in a serum-free medium for 12 h followed by culturing with mediums containing Rol (17772, 5 μg/L, Sigma-Aldrich, St. Louis, MO), atRA (R2625, 5 μmol/L, Sigma-Aldrich, St. Louis, MO), atRA and AGN193109 (SML2034, 1 μmol/L, Sigma-Aldrich, St. Louis, MO) mixture, or equivalent amounts of DMSO (Sigma -Aldrich, St. Louis, MO), respectively. Then, cells were fixed in 4% paraformaldehyde or collected for subsequent analysis 24 h later.

**Drugs preparation.** ROL, atRA, and AGN193109 were first accurately weighed and placed in brown light-proof tubes, then dissolved with DMSO (D8418, Sigma-Aldrich, St. Louis, MO) to prepare working solutions with final concentrations of 5 mg/L, 5 mmol/L, and 2 mmol/L, respectively.

**Oil red O staining.** Cells for oil red O staining were cultured in Millicell EZ SLIDE 8 well glass slides (PEZGS0816, Merck Millipore, Billerica, MA, USA) after isolation, and fixed in 4% paraformaldehyde overnight after treatments. Oil Red O staining was then performed in the same manner as in sections of heart samples.

### Statistics

Statistical differences between groups were evaluated using GraphPad Prism 7 and R 4.2.2. Data were considered statistically significant if $P$ was less than 0.05. All experiments were performed at least in triplicate, and quantitative data are presented as the mean ± SD. Statistical significance for samples was identified using the independent-sample t-test and One-way ANOVA.

### Reporting summary

Further information on research design is available in the Nature Portfolio Reporting Summary linked to this article.

## Data availability

The sequencing data generated in this study have been deposited in the Gene Expression Omnibus under accession code GSE202418. LC-MS data were acquired and analyzed with Analyst® MD version 1.6.3 and Multi Quant TM MD version 3.0.2 (Sciex, M.A., U.S.A) and is available at the NIH Common Fund's National Metabolomics Data Repository (NMDR) website, the Metabolomics Workbench, where it has been assigned Project ID ST002474. The KOBAS (http://kobas.cbi.

pku.edu.cn/) was used for KEGG pathway and GO terms analysis. The ConTra v3 (http://bioit2.irc.ugent.be/contra/v3/#/step/1) was used for RARs binding sites prediction in Supplementary Fig. 8. The remaining data are available within the Article, Supplementary Information, or Source Data file. Source data are provided in this paper. Source data are provided with this paper.

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

## Acknowledgements

We thank Prof. Jianxing Ma (Health Sciences Center, University of Oklahoma) for providing us with RDH10 homozygous floxed mice. We thank Wenshan Zhuo from Instrument Analysis& Research Center, Sun Yat-sen University for the assistance with HPLC analysis in RDHs activity assay. W.B.C is supported by National Key Research and Development Program of China (Grant No.2019YFA0801403), National Nature Science Foundation of China (Grant No. 82170261, 81741117, 81970219), Guangdong Basic and Applied Basic Research Foundation (Grant Number: 2021A1515011005, 2021A1515110233 and 2021B1212040006), Guangzhou Municipal Science and Technology Bureau (Grant Number: 2019030015). X.H.L is supported by National Nature Science Foundation of China (Grant Number. 82000250) and China Postdoctoral Science Foundation (Grant Number: 2020M672976). The funders had no role in the study design, data collection, analysis, decision to publish, or preparation of the manuscript.

## Author contributions

W.B.C and J.D.C conceived, and directed the study. Y.D.W, T.S.H, and X.H.L designed, performed, and analyzed most of the experiments. H.L.R analyzed most of the data. C.H.S, H.P.W, T.W, X.L.F, and S.J.D performed some of the in vitro experiments. Z.Q.F, S.J.X, H.L, S.F.G, Z.Y.Y, F.G and L.L.D performed some of the in vivo experiments. Y.D.W, T.S.H, and X.H.L wrote the manuscript. All authors read and approved the final manuscript. The order of the co-first authors was assigned based on the relative contributions of these individuals.

## Competing interests

The authors declare no competing interests.
