## [Peer Review File · Nature Communications]

Retinol Dehydrogenase 10 Reduction Mediated Retinol Metabolism Disorder Promotes Diabetic CardiomyopathyREVIEWER COMMENTS

Reviewer #1 (Remarks to the Author):

In this manuscript, Wu et al. report that leptin-resistant (db/db) mice display dysregulated cardiac retinoid metabolism that is manifested by increased retinol levels, decreased retinoic acid levels and decreased amount of cardiac retinol dehydrogenase 10 (RDH10). The authors suggest that dysregulation of cardiac retinoid metabolism is a new mechanism underlying diabetic cardiomyopathy. The study incorporates many experiments utilizing different approaches, mouse models as well as human samples.

Potentially, this study could make an important contribution and shed light on the role of retinoid metabolism and, specifically, RDH10 in pathophysiology of diabetic cardiomyopathy. However, a number of methodological deficiencies undermine the significance of reported observations. The major concern is the identity of the protein band that the authors identify as RDH10. In the paper the authors referenced (ref 32), RDH10 protein was barely detectable in 40 µg of microsomes from fasted mouse liver. Wu et al used 35 µg of total protein extract from fasted heart or cardiomyocytes and claim that they are able to detect RDH10 protein band. This is surprising. Furthermore, the quality of westerns raises questions. Some blots show a single band and others- a doublet. In a number of images the upper part of the blot is cut off very close to RDH10 band and molecular weight markers are not indicated on the images. To validate the identity of the band as RDH10, the authors need to provide a clean full size image of the blot with a positive control (recombinant RDH10) to show the size of the RDH10 protein, and include a negative control (sample of their conditional KO total extract) side by side with samples from db/db mice on the same gel/blot. Otherwise, there is a strong possibility that the authors are detecting a non-specific band.

In supplementary Fig 3, the same blot that was incubated with RDH10 antibodies needs to be re-incubated with HSP90 antibodies and the full sized image of the whole blot with two protein bands corresponding to RDH10 and HSP90 on the same blot should be shown to confirm the equal loading. Again, the size markers need to be included on all blots.

Similarly, the IHC results using RDH10 antibodies need to be validated using tissues from RDH10 CKO mice. In general, IHC images are too small, and why is the background blue for IHC in some images (T2D patient in Fig. S2D) but not in others?

Second, to validate the difference in RDH10 protein, the authors need to provide measurements of total RDH activity in microsomal fractions of all samples where RDH10 seems to change.

The authors mention ALDH1A2 and ALDH1A7 as retinoic acid 4 synthesis-related genes. These enzymes are not known to metabolize retinaldehyde to retinoic acid *in vivo*. Likewise, UGT1A10, CYP3A11, UGT1A6B and UGT1A6A are cited as retinoic acid degradation-related genes. References supporting these claims need to be provided.

The result that retinol is increased and atRA is decreased in cardiomyopathy agrees with the previous report from Ni Yang, ... , Maureen A. Kane, D. Brian Foster, JCI Insight. 2021;6(8):e137593.

However, Yang et al. were unable to detect RDH10 in the heart. Hence, it is even more important to prove that antibodies recognize the correct protein corresponding to RDH10 in heart.

The authors show that supplementation of db/db mice with Rol resulted in significantly increased cardiac Rol. This is surprising because retinol administered by oral gavage is delivered for storage in liver and is then distributed to peripheral organs by serum retinol binding protein in a tightly controlled manner. The levels of holo-RBP4 in serum are very constant, so it is unclear how the increase in cardiac retinol would be achieved. Also, excessive retinol is converted to retinyl esters. Have the authors measured retinyl ester levels? Have they examined the levels and activity of LRAT, the enzyme that esterifies retinol?

To demonstrate the role of atRA deficiency in DCM, the authors injected atRA intraperitoneally into db/db mice at 5 µg/g body weight, which amounts to 250 µg/~50 g db/db mouse weight daily for 28 weeks. This dose appeared to restore the atRA in db/db heart to nearly perfectly matched levels to those in control mice. How did the authors determine the exact dose to achieve such a perfect match? Notably, the size of the heart did not appear to change by much based on the images. Compare that to size differences in hearts shown in Ni Yang, ... , Maureen A. Kane, D. Brian Foster, JCI Insight. 2021;6(8):e137593.

To validate the role of RDH10 and cardiac retinol metabolism in DCM, the authors overexpressed

RDH10 in the hearts of db/db mice via adeno-associated virus 9 -RDH10 injection. A nearly perfect match to RDH10 levels in control mice was achieved. How did the authors determine the dose of the virus that gave such a perfect match to control levels? Have they tried a higher dose to increase RDH10 amount further? Did the amount of RDH10 in the heart vary depending on the viral dose? The authors report that pigment epithelium-derived factor (PEDF) binds to RDH10. Considering that RDH10 is ER-bound whereas PEDF is a secretable soluble protein, this finding is surprising and needs to be substantiated by additional experimental evidence.

Calorie restriction was reported to alleviate diabetic cardiomyopathy. Have the authors tried this approach to see if RDH10 levels go up?

Overall, this study has a potential to make an important contribution to the field if the aforementioned concerns are addressed appropriately. English needs editing.

Reviewer #2 (Remarks to the Author):

This study by Wu and colleagues studied the effects of retinol dehydrogenase 10 (RDH10) reduction in diabetic heart as a causative factor for diabetic cardiomyopathy. They show increased cardiac retinol content in the diabetic heart, which is associated with reduced retinoic acid due to the downregulation of the key rate-limiting enzyme RDH10. This is an interesting and novel finding in relation to diabetes and the authors have done a series of experiments to confirm their hypothesis. The authors have provided the full analysis and all blots. My comments are as follows

1. The number of human samples are too small to provide a conclusion or translation value of the study. Importantly, it is not clear on the duration of diabetes in these individuals and whether they were normalized for the intake of medicine and other associated comorbidities. For example, are these patients prescribed any multivitamins? used to conclude the results is very limited.

2. Can the authors please confirm how many times the molecular analysis was repeated independently?

3. Several previous studies have shown the development of systolic and diastolic dysfunction, fibrosis and apoptosis in the db/db mice starting from 20-24 weeks of age. However, in this study, the animals appear to be healthy at least until 24 weeks of age. Please explain the reason for this discrepancy.

4. In line with this, Supplemental figures 2C and 2D show a significant difference between db/db and db/m at 24 weeks of age in EF and FS. However, Figure 2C, where the animals were treated with Rol shows no difference in EF and FS. Not clear why this is different here. Same in Figure 3 – experiments with atRA

5. Similarly, why did the authors choose to follow up to 28 weeks for experiments with atRA. They should have kept the time points consistently across different treatments to make it easy for comparison.

6. Further, no data is presented for 24 weeks time point for the study where they overexpressed RDH10.

Reviewer #3 (Remarks to the Author):

In this study, the authors showed that diabetic cardiomyopathy is associated with increased levels of retinol (Rol), decreased levels of all-trans retinoic acid (atRA), and decreased expression of RDH10 (a key enzyme that mediates the conversion of Rol to atRA). Using overexpression and conditional KO mouse models as well as Rol/atRA supplementation in mice, the authors further showed that dysregulation of Rol metabolism indeed plays a causal role in diabetic cardiomyopathy. They also proposed that deregulation of Rol metabolism affects diabetic cardiomyopathy through several mechanisms, including inducing lipotoxicity and ferroptosis.

Overall, this is a solid study with a lot of in vivo studies and significant pathological implications in diabetic cardiomyopathy. The connection of ferroptosis to diabetic cardiomyopathy is also interesting. However, there are several technical and logical issues in this study that the authors need to thoroughly address.

1. Fig. 2: the authors claim that Rol overload promotes myocardial injury in T2DM mice. However, compared to db/db mice, db/db+Rol mice did not show much difference in terms of heart size and heart/tibia ratio (Fig. 2E), and cardiomyocyte area (Fig. 2G). Therefore, it seems that the data does not justify their conclusion. Note that in these figures, the statistical analysis (which shows the difference is statistically significant) was conducted between db/m and db/db+Rol groups, but this is wrong. Only the comparison between db/db and db/db+Rol groups would make sense here.

2. page 10 "To further identify the regulator of cardiac RDH10 in the heart of T2DM, we treated neonatal mouse primary cardiomyocytes (NMPCs) with high glucose, palmitic acid (PA), recombinant leptin or recombinant pigment epithelium-derived factor (PEDF)". To guide broad readers better here, the authors need to provide a brief justification on why they treated in mice, for example, because these are known factors involved in T2DM?

Likewise, "PEDF, which has been shown to be associated with metabolic diseases 15, 16, 17, is reduced and promotes myocardial injury in the heart in T2DM (unpublished observations)" do the authors mean that the reduction of PEDF expression promotes myocardial injury in the heart in T2DM? The authors need to modify this sentence to make it more clarified.

3. The data Fig. S5E is not consistent with their claim "silencing PEDF enhanced the ubiquitin-mediated degradation of RDH10 in NMPCs (Supplementary Figure 5 E)." There is no difference in polyubiquitin IB between 3rd and 4th lane of RDH10 IP. Also, for Fig. S5B (RDH10 interacts with PEDF), how can PEDF (an extracellular factor) interact with RDH10, which localizes intracellularly?

4. There is a significant disconnection between Fig. S5 and all other figures in this manuscript. PEDF was never mentioned again in the rest of the manuscript, so what is the point to study PEDF in this study? Considering the issues associated with PEDF data (see points 2 and 3) and the lack of data showing that PEDF is important in T2DM (mentioned as unpublished observations. Many journals now do not allow this statement, such that the authors should either show the data or not make the relevant point), I suggest the authors remove Fig. S5 and related text in manuscript. This would not affect the conclusion of this study.

5. Why the authors suddenly jumped to analyze GPX4 in Fig. 7I (considering that there are multiple other ferroptosis regulators)? To make their analyses a bit more unbiased, they should also analyze a few other important ferroptosis suppressors here, including SLC7A11, FSP1, and DHODH (and cite relevant papers. PMID: 22632970, 31634899, 31634900, 33981038).

6. The rescue data by ferroptosis inhibitor in Fig. 7E-H is important. Here they need to analyze a few more parameters in T2DM model (as shown in Fig. E-G).

7. One weakness of this study is that how atRA deficiency causes GPX4 expression suppression, iron accumulation, and CD36-mediated FFA uptake to promote lipotoxicity (as summarized in Fig. 9) remains unknown. Some minimal mechanistic studies can further strengthen the study.

Minor points:

1. The data in Fig. 6J need to be presented with quantification (from at least three data sets) and statistical analyses.

2. Fig. 7A-D, the text states that 4-HNE levels were reduced in RDH10-cKO mice, but the data showed opposite results.

3. Also, the authors mentioned that MDA, iron and non-heme iron levels were not changed in RDH10-cKO mice. Even though ferroptosis is a form of iron-dependent cell death, iron levels per se are not good markers for ferroptosis, so it is not surprising that iron levels did not change here. However, MDA and 4HNE levels typically correlate well with ferroptosis in vivo. Fig 7B indeed shows that MDA levels were increased in RDH10-cKO mice, but the difference is not statistically significant probably because of the huge variation in the analysis. The authors should increase their sample size in their analysis.

**Reviewer #1:**

**In this manuscript, Wu et al. report that leptin-resistant (db/db) mice display**
**dysregulated cardiac retinoid metabolism that is manifested by increased retinol levels,**
**decreased retinoic acid levels and decreased amount of cardiac retinol dehydrogenase 10**
**(RDH10). The authors suggest that dysregulation of cardiac retinoid metabolism is a new**
**mechanism underlying diabetic cardiomyopathy. The study incorporates many**
**experiments utilizing different approaches, mouse models as well as human samples.**

**Potentially, this study could make an important contribution and shed light on the role of**
**retinoid metabolism and, specifically, RDH10 in pathophysiology of diabetic**
**cardiomyopathy. However, a number of methodological deficiencies undermine the**
**significance of reported observations.**

**RESPONSE:** The authors thank the Reviewer for his or her overall supportive comments. At
the same time, we attach great importance to the Reviewer's questions about our study, and we
have conducted additional experiments to answer them point by point in an effort to obtain the
Reviewer's further approval.

**1. The major concern is the identity of the protein band that the authors identify as**
**RDH10. In the paper the authors referenced (ref 32), RDH10 protein was barely**
**detectable in 40 µg of microsomes from fasted mouse liver. Wu et al used 35 µg of total**
**protein extract from fasted heart or cardiomyocytes and claim that they are able to detect**
**RDH10 protein band. This is surprising. Furthermore, the quality of westerns raises**
**questions. Some blots show a single band and others- a doublet. In a number of images**

the upper part of the blot is cut off very close to RDH10 band and molecular weight
markers are not indicated on the images. To validate the identity of the band as RDH10,
the authors need to provide a clean full size image of the blot with a positive control
(recombinant RDH10) to show the size of the RDH10 protein, and include a negative
control (sample of their conditional KO total extract) side by side with samples from
db/db mice on the same gel/blot. Otherwise, there is a strong possibility that the authors
are detecting a non-specific band.

**RESPONSE:** Thank you for your valuable suggestion. We know that your main concern is the
specificity of the RDH10 antibody we used. The RDH10 antibody we used is the same as the
one in reference 32¹; the difference is that in western blotting experiments, we used the
concentration of 1:1000, which is recommended in the manual for this antibody, while the
concentration used in reference 32 was 1:3000.

To further verify the specificity of this antibody, we first followed your suggestion and used
RDH10 recombinant protein as a positive control to revalidate RDH10 expression in *db/db* and
RDH10-cKO mouse hearts. As shown in the image below, the recombinant RDH10 protein is
the same size as the RDH10 protein we detected in our samples. As we showed in the original

1 manuscript, RDH10 expression was significantly downregulated in the hearts of 24- and 32-
2 week-old *db/db* mice.

Second, we compared RDH10 expression in liver microsomes and heart microsomes in fed and
16-h-fasted mice. As shown in the image below, the expression of RDH10 decreased only in
the liver, not in the heart. It is well known that the liver is the major metabolic organ and is
sensitive to metabolic changes, whereas the heart is not as sensitive. As evidence, heart injury
always occurs at the latest stage of metabolic disorders. Therefore, we suggest that starvation
for 16 h or less does not lead to a decrease in RDH10 expression in the hearts of mice.

Third, we repeated all the western blots of RDH10 in Supplementary Figure 3 a and
Supplementary Figure 4 a to address the doublet band of RDH10 issue you pointed out and
showed larger or full size blot images of RDH10 in the source data file to prove the specificity
of our antibody and the reliability of our results.

**2. In supplementary Fig 3, the same blot that was incubated with RDH10 antibodies needs**
**to be re-incubated with HSP90 antibodies and the full-sized image of the whole blot with**

**two protein bands corresponding to RDH10 and HSP90 on the same blot should be shown**
**to confirm the equal loading. Again, the size markers need to be included on all blots.**

**RESPONSE:** Thank you for your valuable advice. First, it needs to be explained that in our
original manuscript, the strips of RDH10 and HSP90 were cropped from the same PVDF
membrane, and there was no difference in the loading. However, in view of the reviewer's query,
we revalidated our results of RDH10 western blotting in Supplementary Figure 4 a and hope
we have alleviated your doubts and can obtain your approval.

**3. Similarly, the IHC results using RDH10 antibodies need to be validated using tissues**
**from RDH10 CKO mice. In general, IHC images are too small, and why is the background**
**blue for IHC in some images (T2D patient in Fig. S2D) but not in others?**

**RESPONSE:** We know this comment reflects the same concern as that for the specificity of
our RDH10 antibody. We have followed your suggestions to verify the specificity of the
antibody we used, as described in the 2 responses above, and we hope our measures will address
your concerns. In addition, we performed IHC staining of RDH10 in RDH10-cKO mice, and
as shown in Supplementary Figure 4 b, cardiac RDH10 expression was decreased in RDH10-
cKO mice, which is consistent with our western blotting and IF staining results.

The background of IHC staining was determined by the time of hematoxylin staining and the
microscope parameters used. We erred in not keeping the microscopy parameters consistent
from experiment to experiment; however, we kept the same parameters of the microscope
within the same experiment (panel), so the background of IHC in the same panel is the same.

When IHC staining is performed, the brown positive DAB staining will cover the purple color
of hematoxylin staining, so not maintaining the experimental microscope parameters should
not have affected the conclusions we obtained in the same batch of experiments.

**4. Second, to validate the difference in RDH10 protein, the authors need to provide**
**measurements of total RDH activity in microsomal fractions of all samples where RDH10**
**seems to change.**

**RESPONSE:** Thank you for your very knowledgeable advice. Based on your suggestion, we
assayed the activity of RDHs in the hearts of 32 w *db/db* mice. As shown in Supplementary
Figure 3 c, the activity of RDHs was significantly decreased in the hearts of *db/db* mice during
heart failure.

**5. The authors mention ALDH1A2 (RALDH2) and ALDH1A7 as retinoic acid 4**
**synthesis-related genes. These enzymes are not known to metabolize retinaldehyde to**
**retinoic acid in vivo. Likewise, UGT1A10, CYP3A11, UGT1A6B (UGT1A9) and**
**UGT1A6A (UDPGT, UGT1A7) are cited as retinoic acid degradation-related genes.**
**References supporting these claims need to be provided.**

**RESPONSE:** Thank you for pointing out this issue. As shown in [https://www.genome.jp/kegg-](https://www.genome.jp/kegg-bin/show_pathway?mmu00830)
[bin/show_pathway?mmu00830](https://www.genome.jp/kegg-bin/show_pathway?mmu00830) and the following pictures, the above genes were classified into
the retinoic acid synthesis and retinoic acid degradation pathways by KEGG pathway analysis.
However, we did not find any papers supporting the idea that UGT1A10 is associated with
retinoic acid degradation, so we have removed UGT1A10 from the corresponding position in

- 1 the manuscript and Figure 1. For the rest of the genes, we found literature that supports the
- 2 findings, and this literature is cited in the manuscript.

- 1 6. The result that retinol is increased and atRA is decreased in cardiomyopathy agrees
- 2 with the previous report from Ni Yang, ... , Mauren A. Kane, D. Brian Foster, JCI

**Insight. 2021;6(8):e137593. However, Yang et al. were unable to detect RDH10 in the**
**heart. Hence, it is even more important to prove that antibodies recognize the correct**
**protein corresponding to RDH10 in heart.**

**RESPONSE:** Thank you for your comment. We read the reference you mentioned (reference
9) again carefully and did not find results of RDH10 western blotting in that paper. The authors
of that reference only mentioned that they did not find RDH10 in their proteomics data and
explained that “One of the best-characterized RDHs, Rdh10, was not detected. However, since
our iTRAQ proteomics workflow employed data-dependent peptide acquisition, the absence of
evidence for Rdh10 should not necessarily be interpreted as evidence of its absence”². As the
authors said, iTRAQ proteomics is not sensitive enough to detect all the proteins in the tissues.
In fact, we have also tried to find the potential mechanism of diabetic cardiomyopathy via
iTRAQ proteomics; however, we only identified 3968 proteins in the hearts of mice by this
technique, which is far less than the total number of proteins present in the heart. Therefore, the
absence of RDH10 in the heart detected by iTRAQ proteomics does not mean that RDH10 is
not present in the heart.

**7. The authors show that supplementation of *db/db* mice with Rol resulted in significantly**
**increased cardiac Rol. This is surprising because retinol administered by oral gavage is**
**delivered for storage in liver and is then distributed to peripheral organs by serum retinol**
**binding protein in a tightly controlled manner. The levels of holo-RBP4 in serum are very**
**constant, so it is unclear how the increase in cardiac retinol would be achieved. Also,**
**excessive retinol is converted to retinyl esters. Have the authors measured retinyl ester**

**levels? Have they examined the levels and activity of LRAT, the enzyme that esterifies**
**retinol?**

**RESPONSE:** Thank you for your knowledgeable advice. Retinol administration by oral gavage
is common in related studies ^{3, 4}, and the dose of Rol we used was converted from the highest
dose that humans can use as a supplement without toxicity. We have previously attempted to
use higher doses, but they produced significant toxic effects in mice.

Numerous studies have found that dietary supplementation with Rol or its prerequisites can
significantly elevate Rol concentrations in the blood, liver, lungs and kidneys of mice, ducks
and humans ^{5, 6, 7}. However, we did not find any study suggesting that long-term Rol
supplementation does not alter serum holo-RBP4 levels. It is well known that skin toxicity and
neurotoxicity, as well as very significant teratogenic effects, occur if Rol is supplemented in
excess ⁸. We believe that these phenomena cannot be caused solely by excessive accumulation
of Rol in the liver.

Due to the lack of a validated assay for blood holo-RBP4 levels in mice and to verify that Rol
supplementation elevates Rol levels in serum and other non-liver tissues, we measured the
RBP4 levels in serum and Rol levels in both the serum and hearts of *db/db* mice supplied with
Rol for 2 months (800 IU/2 days). Our results showed that supplementation with Rol over a
relatively short period significantly increased cardiac Rol levels but did not change serum RBP4
and Rol levels in these mice. This result surprised us, and it also made us realize that there are
still many unanswered questions in the field of retinol metabolism. We are unable to explain

this phenomenon, but we believe that through concerted efforts, researchers will be able to
solve the mysteries of retinol metabolism.

Because of Covid-19, we were unable to obtain a suitable antibody to measure the protein
expression level of LRAT within the revision period, we only measured the concentration of
cardiac retinyl esters, which were not altered in either *db/db* or *db/db+Rol* mice as shown in
Supplementary Figure 2.

**8. To demonstrate the role of atRA deficiency in DCM, the authors injected atRA**
**intraperitoneally into *db/db* mice at 5 μ g/g body weight, which amounts to 250 μ g/~50 g**
***db/db* mouse weight daily for 28 weeks. This dose appeared to restore the atRA in *db/db***
**heart to nearly perfectly matched levels to those in control mice. How did the authors**
**determine the exact dose to achieve such a perfect match?**

**RESPONSE:** The way we provided atRA to *db/db* mice and the dose used were selected by
 referring to a relevant study ⁹. In the study we referenced, atRA served the purpose of treating
 myocardial injury, but perhaps due to the difficulty of atRA measurement, the authors did not
 measure the cardiac atRA levels in the mice before and after atRA supplementation. Due to the
 extremely unstable physicochemical properties of atRA, measurement of atRA levels in tissue
 is very difficult, so we specifically found a partner to help us measure cardiac atRA levels by
 establishing a very reliable method. We did not anticipate that such a dose of atRA could so
 perfectly restore the cardiac atRA content in *db/db* mice to about the same level as in *db/m* mice,
 but it did. We have attached the original peak plots of the cardiac atRA content we performed
 for your review and hope they will meet with your approval.

**9. Notably, the size of the heart did not appear to change by much based on the images.**
 **Compare that to size differences in hearts shown in Ni Yang, ... , Maureen A. Kane, D.**
 **Brian Foster, JCI Insight. 2021;6(8):e137593.**

**RESPONSE:** Thank you for your comments. As shown in Figure 3 e, atRA supplementation
 reduced the heart weights of *db/db* mice, as indicated in images and by statistical analysis, a
 phenomenon identical to that in the work of Ni Yang et al.

**10.To validate the role of RDH10 and cardiac retinol metabolism in DCM, the authors**
**overexpressed RDH10 in the hearts of db/db mice via adeno-associated virus 9 -RDH10**
**injection. A nearly perfect match to RDH10 levels in control mice was achieved. How did**
**the authors determine the dose of the virus that gave such a perfect match to control levels?**
**Have they tried a higher dose to increase RDH10 amount further? Did the amount of**
**RDH10 in the heart vary depending on the viral dose?**

**RESPONSE:** Thank you for your comments. In fact, we performed preliminary experiments
using three concentrations of AAV9 virus, 0.5×10^{11} , 0.8×10^{11} and 1.2×10^{11} . We found
that 0.5×10^{11} of AAV9 virus did not achieve the desired overexpression effect, while
1.2×10^{12} of AAV9 virus caused the mice to die one after another in one month (including in
the AAV9-RDH10 group and AAV9-GFP group), which we think may have been caused by
the toxicity of AAV9 itself.

In addition, the aim of our experiment was to restore cardiac RDH10 expression in *db/db* mice
to the level in *db/m* mice rather than to achieve infinite overexpression of cardiac RDH10 in
*db/db* mice, so we ultimately chose a virus dose of 0.8×10^{11} . As shown in the figure below,
there were individual differences among different mice, so we could not accurately restore
cardiac RDH10 of each mouse to a level comparable to that in *db/m* mice. In Supplementary

Figure 5, we show the mice from the AAV9-RDH10 group with the cardiac RDH10 levels
closest to those of *db/m* mice.

**11. The authors report that pigment epithelium-derived factor (PEDF) binds to RDH10.**
**Considering that RDH10 is ER-bound whereas PEDF is a secretable soluble protein, this**
**finding is surprising and needs to be substantiated by additional experimental evidence.**

**RESPONSE:** Thank you for your comment. PEDF is a secretable protein that was once thought
to be produced only in the liver and adipose tissue, but as research has progressed, an increasing
number of researchers have found that cardiomyocytes can also produce PEDF^{10, 11, 12, 13}. We
tried to verify the possibility that RDH10 binds to PEDF in cardiomyocytes using
immunofluorescence colocalization, but since we were not able to find an antibody that could
be used for mouse heart immunofluorescence staining for PEDF, we could only demonstrate
that PEDF was indeed present in the mouse myocardium by IHC staining and that PEDF
expression was significantly reduced in the hearts of *db/db* mice.

In view of reviewer #3's suggestion that we should not cite data that are not yet published, we
have removed all PEDF-related results from the manuscript and placed them in our forthcoming
submission on PEDF involvement in diabetic cardiomyopathy.

**12. Calorie restriction was reported to alleviate diabetic cardiomyopathy. Have the**
**authors tried this approach to see if RDH10 levels go up?**

**RESPONSE:** Thank you for the wonderful idea. The idea you mentioned is very interesting
and a very worthy subject to be studied in depth, but due to the time limit of the revision period,
we could not finish this study in a limited time. The possibility you propose is a profound
inspiration to us, and we will study and explore this subject in depth in the future. Thank you
again.

**13. Overall, this study has a potential to make an important contribution to the field if the**
**aforementioned concerns are addressed appropriately.**

**RESPONSE:** We thank you for your recognition of this work and hope that our answers will
address your concerns to your satisfaction. We also hope that this work will make some
contribution to the related field and promote its further development.

**Reviewer #2:**

**This study by Wu and colleagues studied the effects of retinol dehydrogenase 10 (RDH10)**
**reduction in diabetic heart as a causative factor for diabetic cardiomyopathy. They show**
**increased cardiac retinol content in the diabetic heart, which is associated with reduced**
**retinoic acid due to the downregulation of the key rate-limiting enzyme RDH10. This is**
**an interesting and novel finding in relation to diabetes and the authors have done a series**

**of experiments to confirm their hypothesis. The authors have provided the full analysis**
**and all blots.**

**RESPONSE:** The authors thank the Reviewer for his or her overall supportive comments. At
the same time, we attach great importance to the Reviewer's questions about our study. We
have conducted additional experiments to answer them point by point and hope to obtain the
Reviewer's further approval.

**1. The number of human samples are too small to provide a conclusion or translation**
**value of the study. Importantly, it is not clear on the duration of diabetes in these**
**individuals and whether they were normalized for the intake of medicine and other**
**associated comorbidities. For example, are these patients prescribed any multivitamins?**
**used to conclude the results is very limited.**

**RESPONSE:** Your comments are critical, and this is indeed a major deficiency in our study.
Human heart samples are very difficult to obtain, and this is a common challenge for researchers
in the field. However, to further increase the reliability of our study, we have done our best to
collect additional human samples and have updated our relevant results in Figure 1 h. We have
also summarized and presented information about these samples to the extent possible in
Supplementary Table 3, but unfortunately, we were unable to obtain information on whether
these patients had taken a multivitamin.

**2. Can the authors please confirm how many times the molecular analysis was repeated**
**independently?**

**RESPONSE:** Thank you for your suggestion. We have carefully confirmed the replicates of
all experiments and clearly marked them in the Figure Legends. All of our experimental
replicates meet the statistical requirements.

**3. Several previous studies have shown the development of systolic and diastolic**
**dysfunction, fibrosis and apoptosis in the db/db mice starting from 20-24 weeks of age.**
**However, in this study, the animals appear to be healthy at least until 24 weeks of age.**
**Please explain the reason for this discrepancy.**

**RESPONSE:** Thank you for your question. It is inevitable that there is some individual
variation among different batches of *db/db* mice, and publications by different groups have
shown different times of heart systolic dysfunction in *db/db* mice^{14, 15, 16, 17}. Our group has also
made long-term observations on the heart function of *db/db* mice and found that heart systolic
dysfunction does not appear in *db/db* mice until 32 weeks of age. The relevant data were
published as a cover article in *Theranostics* in 2022¹⁶.

**4. In line with this, Supplemental figures 2C and 2D show a significant difference between**
**db/db and db/m at 24 weeks of age in EF and FS. However, Figure 2C, where the animals**
**were treated with Rol shows no difference in EF and FS. Not clear why this is different**
**here. Same in Figure 3 – experiments with atRA.**

**RESPONSE:** Thank you for your question. As our response above states, it is inevitable that
there is some individual variation among different batches of *db/db* mice; the *db/db* mice we
used in Supplementary Figure 2 c are not from the same batch as the *db/db* mice we used in
Figure 3. In addition, during this study period, we acquired the latest small animal ultrasound

detector, a Vevo 3100, by ourselves; only the data in Supplementary Figure 2 were detected
using a slightly earlier version of small animal ultrasound detector, the Vevo 2100, from others
before we acquired this instrument. We believe that different batches of mice and the different
small animal ultrasound instruments may have contributed to the differences in Supplementary
Figure 2 and others. However, our results in the same figure are based on data from the same
batch of animals with the same animal ultrasound detector, so the above differences should not
affect the reliability of our results.

**5. Similarly, why did the authors choose to follow up to 28 weeks for experiments with**
**atRA. They should have kept the time points consistently across different treatments to**
**make it easy for comparison.**

**RESPONSE:** Thank you for the knowledgeable comment. Normally, we would set the same
observation time point for both groups, but the *db/db*+Rol mice in the Rol supplementation
experiment group already showed significant heart systolic dysfunction at 24 w. If we continued
to extend the observation time, it may have led to a large number of deaths in the *db/db*+Rol
mice, which would not have been conducive to our data collection and analysis. Therefore, we
chose 24 was the experimental endpoint in this part of the experiment. In the atRA
supplementation experiment group, no mice showed significant heart systolic dysfunction at 24
w. It was not possible to determine the effect of atRA supplementation on heart function in
*db/db* mice at 24 w, so we continued the test until 28 w, when the *db/db* mice without atRA
supplementation showed significant heart systolic dysfunction.

**6. Further, no data is presented for 24 weeks time point for the study where they**
**overexpressed RDH10.**

**RESPONSE:** Thank you for the very helpful comment. We did have this flaw in our process
of experimentation. We chose to only measure regularly and missed the measurement at 24 w,
but the data at 28 w are sufficient to show the protective effects of AAV9-RDH10 on heart
structure and function in *db/db* mice, so the lack of 24 w data does not affect our final
experimental conclusion.

**Reviewer #3:**

**In this study, the authors showed that diabetic cardiomyopathy is associated with**
**increased levels of retinol (Rol), decreased levels of all-trans retinoic acid (atRA), and**
**decreased expression of RDH10 (a key enzyme that mediates the conversion of Rol to**
**atRA). Using overexpression and conditional KO mouse models as well as Rol/atRA**
**supplementation in mice, the authors further showed that dysregulation of Rol**
**metabolism indeed plays a causal role in diabetic cardiomyopathy. They also proposed**
**that deregulation of Rol metabolism affects diabetic cardiomyopathy through several**
**mechanisms, including inducing lipotoxicity and ferroptosis.**

**Overall, this is a solid study with a lot of in vivo studies and significant pathological**
**implications in diabetic cardiomyopathy. The connection of ferroptosis to diabetic**
**cardiomyopathy is also interesting. However, there are several technical and logical issues**
**in this study that the authors need to thoroughly address.**

**RESPONSE:** The authors thank the Reviewer for his or her overall supportive comments. At
the same time, we attach great importance to the Reviewer's questions about our study. We
have conducted additional experiments to answer them point by point and hope to obtain the
Reviewer's further approval.

**1. Fig. 2: the authors claim that Rol overload promotes myocardial injury in T2DM mice.**
**However, compared to db/db mice, db/db+Rol mice did not show much difference in**
**terms of heart size and heart/tibia ratio (Fig. 2E), and cardiomyocyte area (Fig. 2G).**
**Therefore, it seems that the data does not justify their conclusion. Note that in these**
**figures, the statistical analysis (which shows the difference is statistically significant) was**
**conducted between db/m and db/db+Rol groups, but this is wrong. Only the comparison**
**between db/db and db/db+Rol groups would make sense here.**

**RESPONSE:** Thank you for your knowledgeable comment. Based on your suggestion, we
double-checked our statistics and those in the figure you mentioned (Figure 2). In fact, we
compared db/db+Rol mice with db/m and db/db mice and have specified this in the graph and
manuscript. Our results showed no significant difference between the db/db+Rol group and the
db/db group, which is why we only identified the p value between the db/db+Rol group and the
db/m group in Figure 2.

Furthermore, in the figure you mentioned (Figure 2), although db/db+Rol mice did not show
much difference in heart size and heart/tibial ratio compared to db/db mice (Figure 2 e and g),
they showed significant differences in heart function, myocardial fibrosis, and apoptosis in the
myocardium (Figure 2 b, c, f and g). In conclusion, db/db+Rol mice exhibit heart systolic

dysfunction earlier and with more severe myocardial fibrosis and apoptosis than db/db mice.
These findings are sufficient to suggest that Rol supplementation impairs heart structure and
function in db/db mice, but not through myocardial hypertrophy.

**2. page 10 “To further identify the regulator of cardiac RDH10 in the heart of T2DM, we**
**treated neonatal mouse primary cardiomyocytes (NMPCs) with high glucose, palmitic**
**acid (PA), recombinant leptin or recombinant pigment epithelium-derived factor**
**(PEDF)”. To guide broad readers better here, the authors need to provide a brief**
**justification on why they treated in mice, for example, because these are known factors**
**involved in T2DM?**

**Likewise, “PEDF, which has been shown to be associated with metabolic diseases 15, 16,**
**17, is reduced and promotes myocardial injury in the heart in T2DM (unpublished**
**observations)” do the authors mean that the reduction of PEDF expression promotes**
**myocardial injury in the heart in T2DM? The authors need to modify this sentence to**
**make it more clarified.**

**RESPONSE:** Thank you for your knowledgeable advice. This part of the experiment was
designed to explore the factors that may lead to a decrease in the cardiac RDH10 level in
diabetes. Among the stimuli used, glucose, palmitic acid and leptin are all known inducers of
myocardial damage in diabetes^{18, 19}. PEDF is a protective agent against diabetic cardiomyopathy
that has been demonstrated in another of our studies. To prove the authenticity of our study, we
have provided some of the relevant data for your reference. In view of your suggestion #4, we
decided to remove the content about PEDF from this manuscript and will further refine and
enrich that information to add to our related study on PEDF and diabetic cardiomyopathy.

**3. The data Fig. S5E is not consistent with their claim “silencing PEDF enhanced the**
 **ubiquitin-mediated degradation of RDH10 in NMPCs (Supplementary Figure 5 E).”**
 **There is no difference in polyubiquitin IB between 3rd and 4th lane of RDH10 IP. Also,**
 **for Fig. S5B (RDH10 interacts with PEDF), how can PEDF (an extracellular factor)**
 **interact with RDH10, which localizes intracellularly?**
 **RESPONSE:** Thank you for your careful and helpful advice. We have repeated the experiment,
 and the results show a clearer regulation of ubiquitination of RDH10 by PEDF.

PEDF was once thought to be a secretory protein secreted only by the liver and fat, but an
 increasing number of studies have found that PEDF is also expressed in the heart and
 cardiomyocytes^{10, 11, 12, 13}. We did not find suitable PEDF antibodies to verify the colocalization
 of PEDF and RDH10 in mouse cardiomyocytes, but we verified by IHC staining that PEDF is
 indeed abundantly expressed in the mouse heart.

**4. There is a significant disconnection between Fig. S5 and all other figures in this**
 **manuscript. PEDF was never mentioned again in the rest of the manuscript, so what is**
 **the point to study PEDF in this study? Considering the issues associated with PEDF data**
 **(see points 2 and 3) and the lack of data showing that PEDF is important in T2DM**

(mentioned as unpublished observations. Many journals now do not allow this statement,
such that the authors should either show the data or not make the relevant point), I
suggest the authors remove Fig. S5 and related text in manuscript. This would not affect
the conclusion of this study.

**RESPONSE:** Thank you for the professional advice. We have deleted Fig. S5, and we will add
the results in this figure to our next study on PEDF and diabetic cardiomyopathy.

**5. Why the authors suddenly jumped to analyze GPX4 in Fig. 7I (considering that there
are multiple other ferroptosis regulators)? To make their analyses a bit more unbiased,
they should also analyze a few other important ferroptosis suppressors here, including
SLC7A11, FSP1, and DHODH (and cite relevant papers. PMID: 22632970, 31634899,
31634900, 33981038).**

**RESPONSE:** Thank you for your knowledgeable advice. The detection of GPX4 only was
indeed a major shortcoming in our study. As you suggested, we have added detection of
SLC7A11, FSP1, DHODH, and Fpn, as shown in Figure 7, Figure 8 and Supplementary Figure
6. In addition to GPX4, FSP1 and Fpn contribute to the development of retinol metabolism
disorder-induced heart injury in T2DM.

**6. The rescue data by ferroptosis inhibitor in Fig. 7E-H is important. Here they need to
analyze a few more parameters in T2DM model (as shown in Fig. E-G).**

**RESPONSE:** Thank you for your knowledgeable advice. The relationship between ferroptosis
and DCM is indeed a very interesting and attention-grabbing topic. However, complementary
experiments to demonstrate the effect of Fer-1 on heart structure and function in *db/db* mice

will take a very long time, and we could not accomplish this work in the short revision period
of 6 months. After a careful review of the literature, we found that during our preparation and
submission of this manuscript, two studies have already verified the role of ferroptosis in DCM
in T2DM. They have validated the cardioprotective effects of Fer-1 in *db/db* mice and
liproxstatin-1 in mice with STZ-induced diabetic mice, respectively^{17, 20}, demonstrating that
ferroptosis is a key factor in the development of DCM in T2DM. In our study, we concluded
that ferroptosis is involved in DCM by further analysis of RNA- seq data and detection of
ferroptosis markers in the hearts of *db/db* mice and T2DM patients likewise. In response to the
two publications, we have made additional changes to the relevant content of our manuscript.
Additionally, in this study, to avoid repeating work already done by others, we no longer
investigated the cardioprotective effect of Fer-1 on T2DM mice but focused on evidence that
disorders of retinol metabolism are the key initiating factors for the appearance of ferroptosis
in the heart in T2DM.

**7. One weakness of this study is that how atRA deficiency causes GPX4 expression**
**suppression, iron accumulation, and CD36-mediated FFA uptake to promote lipotoxicity**
**(as summarized in Fig. 9) remains unknown. Some minimal mechanistic studies can**
**further strengthen the study.**

**RESPONSE:** Thank you for your knowledgeable comments. First, it has been demonstrated in
atherosclerosis that synthetic retinoic acid downregulates CD36 by inhibiting interleukin 6
expression²¹, and it was on the basis of this finding that we verified that retinoic acid also
inhibits myocardial CD36 expression in DCM. Then, there are no studies on the relationship
between GPX4 and retinoid metabolism, and as shown in Supplementary Figure 8, our analysis

revealed that there are many binding sites for RARs on the promoter of GPX4. RARs are
members of the nuclear receptor family and can regulate the transcription of target genes when
activated by atRA, so we further verified and demonstrated the regulation of GPX4 transcript
levels by retinoic acid and retinoic acid receptors. Based on the above results, we suggest that
retinol metabolism due to RDH10 deficiency reduces GPX4 expression by causing a lack of
atRA-RARs signaling, which inhibits GPX4 transcription.

**Minor points:**

**1. The data in Fig. 6J need to be presented with quantification (from at least three data**
**sets) and statistical analyses.**

**RESPONSE:** Thank you for the comments. Following your suggestion, we have added the
repeats and statistics for the data in Figure 6 j and hope to obtain your approval.

**2. Fig. 7A-D, the text states that 4-HNE levels were reduced in RDH10-cKO mice, but the**
**data showed opposite results.**

**RESPONSE:** Thank you for your careful reading and comments. We apologize for the mistake
in writing and have corrected it in the manuscript. The levels of 4-HNE were increased in the
hearts of RDH10-cKO mice.

**3. Also, the authors mentioned that MDA, iron and non-heme iron levels were not changed**
**in RDH10-cKO mice. Even though ferroptosis is a form of iron-dependent cell death, iron**
**levels per se are not good markers for ferroptosis, so it is not surprising that iron levels**
**did not change here. However, MDA and 4HNE levels typically correlate well with**
**ferroptosis in vivo. Fig 7B indeed shows that MDA levels were increased in RDH10-cKO**

mice, but the difference is not statistically significant probably because of the huge
variation in the analysis. The authors should increase their sample size in their analysis.

**RESPONSE:** Thank you for your expert advice. We increased the number of RDH10-cKO
mice we used and found that as shown in Figure 7, the expanded sample size resulted in
significant differences in cardiac MDA levels between RDH10^{fl/fl} and RDH10-cKO mice, while
cardiac iron and cardiac nonheme iron levels remained unchanged.

**References**

- 1. Klyuyeva AV, Belyaeva OV, Goggans KR, Krezel W, Popov KM, Kedishvili NY. Changes in retinoid
metabolism and signaling associated with metabolic remodeling during fasting and in type I diabetes.
*The Journal of biological chemistry* **296**, 100323 (2021).
- 2. Yang N, *et al.* Cardiac retinoic acid levels decline in heart failure. *JCI insight* **6**, (2021).
- 3. de Oliveira MR, de Bittencourt Pasquali MA, Silvestrin RB, Mello EST, Moreira JC. Vitamin A
supplementation induces a prooxidative state in the striatum and impairs locomotory and exploratory
activity of adult rats. *Brain research* **1169**, 112-119 (2007).
- 4. Afshari-Kaveh M, Abbasalipourkabir R, Nourian A, Ziamajidi N. The Protective Effects of Vitamins A
and E on Titanium Dioxide Nanoparticles (nTiO₂)-Induced Oxidative Stress in the Spleen Tissues of
Male Wistar Rats. *Biological trace element research* **199**, 3677-3687 (2021).
- 5. Penkert RR, *et al.* Vitamin A Corrects Tissue Deficits in Diet-Induced Obese Mice and Reduces Influenza
Infection After Vaccination and Challenge. *Obesity (Silver Spring, Md)* **28**, 1631-1636 (2020).
- 6. Feng YL, Xie M, Tang J, Huang W, Zhang Q, Hou SS. Effects of vitamin A on growth performance and
tissue retinol of starter White Pekin ducks. *Poultry science* **98**, 2189-2192 (2019).
- 7. Liu L, Tang XH, Gudas LJ. Homeostasis of retinol in lecithin: retinol acyltransferase gene knockout mice
fed a high retinol diet. *Biochemical pharmacology* **75**, 2316-2324 (2008).
- 8. Penniston KL, Tanumihardjo SA. The acute and chronic toxic effects of vitamin A. *The American journal*
*of clinical nutrition* **83**, 191-201 (2006).
- 9. Park SW, Nhieu J, Lin YW, Wei LN. All-trans retinoic acid attenuates isoproterenol-induced cardiac
dysfunction through Crabp1 to dampen CaMKII activation. *European journal of pharmacology* **858**,
172485 (2019).

10. Crowe S, *et al.* Pigment epithelium-derived factor contributes to insulin resistance in obesity. *Cell*
*metabolism* **10**, 40-47 (2009).
11. Huang KT, Lin CC, Tsai MC, Chen KD, Chiu KW. Pigment epithelium-derived factor in lipid metabolic
disorders. *Biomedical journal* **41**, 102-108 (2018).
12. Zhang H, *et al.* PEDF improves cardiac function in rats with acute myocardial infarction via inhibiting
vascular permeability and cardiomyocyte apoptosis. *International journal of molecular sciences* **16**,
5618-5634 (2015).
13. Zhang H, *et al.* PEDF and 34-mer inhibit angiogenesis in the heart by inducing tip cells apoptosis via up-
regulating PPAR- γ to increase surface FasL. *Apoptosis : an international journal on programmed cell*
*death* **21**, 60-68 (2016).
14. Arow M, *et al.* Sodium-glucose cotransporter 2 inhibitor Dapagliflozin attenuates diabetic
cardiomyopathy. *Cardiovascular diabetology* **19**, 7 (2020).
15. Yin Z, *et al.* MiR-30c/PGC-1 β protects against diabetic cardiomyopathy via PPAR α . *Cardiovascular*
*diabetology* **18**, 7 (2019).
16. Li X, *et al.* Distinct cardiac energy metabolism and oxidative stress adaptations between obese and non-
obese type 2 diabetes mellitus. *Theranostics* **10**, 2675-2695 (2020).
17. Ni T, Huang X, Pan S, Lu Z. Inhibition of the long non-coding RNA ZFAS1 attenuates ferroptosis by
sponging miR-150-5p and activates CCND2 against diabetic cardiomyopathy. *Journal of cellular and*
*molecular medicine* **25**, 9995-10007 (2021).
18. Liu F, *et al.* Upregulation of MG53 induces diabetic cardiomyopathy through transcriptional activation
of peroxisome proliferation-activated receptor α . *Circulation* **131**, 795-804 (2015).
19. Li K, *et al.* Tetrahydrocurcumin Ameliorates Diabetic Cardiomyopathy by Attenuating High Glucose-
Induced Oxidative Stress and Fibrosis via Activating the SIRT1 Pathway. *Oxidative medicine and*
*cellular longevity* **2019**, 6746907 (2019).
20. Wang X, *et al.* Ferroptosis is essential for diabetic cardiomyopathy and is prevented by sulforaphane via
AMPK/NRF2 pathways. *Acta pharmaceutica Sinica B* **12**, 708-722 (2022).
21. Takeda N, *et al.* Synthetic retinoid Am80 reduces scavenger receptor expression and atherosclerosis in
mice by inhibiting IL-6. *Arteriosclerosis, thrombosis, and vascular biology* **26**, 1177-1183 (2006).

REVIEWERS' COMMENTS

Reviewer #1 (Remarks to the Author):

The authors did a good job addressing my comments. I only have one more suggestion. In the abstract, the authors state: "...demonstrated that both cardiac retinol overload and all-trans retinoic acid deficiency promote diabetic cardiomyopathy by supplementing type 2 diabetic mice with retinol or all-trans retinoic acid." This is confusing because as stated, the sentence implies that supplementing T2D mice with retinol and ATRA promotes diabetic cardiomyopathy. In fact, the authors are saying the opposite. Please rephrase. Perhaps, like this: "By supplementing type 2 diabetic mice with retinol or all-trans retinoic acid, we demonstrate that both cardiac retinol overload and all-trans retinoic acid deficiency promote diabetic cardiomyopathy."

Reviewer #3 (Remarks to the Author):

The authors have addressed the comments from this reviewer. The manuscript can be accepted for its publication in Nature Communications.

Reviewer #4 (Remarks to the Author):

Lu Cai

Reviewer #1:

Common: The authors did a good job addressing my comments. I only have one more suggestion. In the abstract, the authors state: "...demonstrated that both cardiac retinol overload and all-trans retinoic acid deficiency promote diabetic cardiomyopathy by supplementing type 2 diabetic mice with retinol or all-trans retinoic acid." This is confusing because as stated, the sentence implies that supplementing T2D mice with retinol and ATRA promotes diabetic cardiomyopathy. In fact, the authors are saying the opposite. Please rephrase. Perhaps, like this: "By supplementing type 2 diabetic mice with retinol or all-trans retinoic acid, we demonstrate that both cardiac retinol overload and all-trans retinoic acid deficiency promote diabetic cardiomyopathy."

RESPONSE: Thank you for your careful observation and consideration. According to your suggestion, we have made the corresponding changes in the Abstract, and hope to get your approval.

Reviewer #3:

Common: The authors have addressed the comments from this reviewer. The manuscript can be accepted for its publication in Nature Communications.

RESPONSE: Thank you for your affirmation, your suggestions have helped us to further improve our work.

Reviewer #4:

Common: Lu Cai

RESPONSE: Thank you for your affirmation.